# Evolutionarily conserved regulation of immunity by the splicing factor RNP-6/PUF60

**Chun Kew[1†], Wenming Huang[1†], Julia Fischer[2,3,4,5], Raja Ganesan[6], Nirmal Robinson[6], Adam Antebi[1,7]***

[1]Max Planck Institute for Biology of Ageing, Cologne, Germany; [2]Department I of Internal Medicine, University of Cologne, Cologne, Germany; [3]Division of Infectious Diseases, University of Cologne, Cologne, Germany; [4]German Center for Infection Research (DZIF), Partner Site Bonn-Cologne, Cologne, Germany; [5]Center for Molecular Medicine Cologne (CMMC), Cologne, Germany; [6]Cellular-Stress and Immune Response Laboratory, Centre for Cancer Biology, University of South Australia, Adelaide, Australia; [7]Cologne Excellence Cluster on Cellular Stress Responses in Aging-Associated Diseases (CECAD), University of Cologne, Cologne, Germany

*For correspondence: antebi@age.mpg.de

[†]These authors contributed equally to this work

Competing interests: The authors declare that no competing interests exist.

**Abstract** Splicing is a vital cellular process that modulates important aspects of animal physiology, yet roles in regulating innate immunity are relatively unexplored. From genetic screens in *C. elegans*, we identified splicing factor RNP-6/PUF60 whose activity suppresses immunity, but promotes longevity, suggesting a tradeoff between these processes. Bacterial pathogen exposure affects gene expression and splicing in a *rnp-6* dependent manner, and *rnp-6* gain and loss-of-function activities reveal an active role in immune regulation. Another longevity promoting splicing factor, SFA-1, similarly exerts an immuno-suppressive effect, working downstream or parallel to RNP-6. RNP-6 acts through TIR-1/PMK-1/MAPK signaling to modulate immunity. The mammalian homolog, PUF60, also displays anti-inflammatory properties, and its levels swiftly decrease after bacterial infection in mammalian cells, implying a role in the host response. Altogether our findings demonstrate an evolutionarily conserved modulation of immunity by specific components of the splicing machinery.

## Introduction

Innate immunity comprises the first line of defense against invading micro-organisms. The presence of pathogen associated molecular patterns (PAMPs) or damage associated molecular patterns (DAMPs) activates conserved signaling pathways to protect the host and bring infection under control (*Zindel and Kubes, 2020*). Over the last several decades, several powerful invertebrate models including the round worm *Caenorhabditis elegans* have provided significant and unique insights into underlying mechanisms of innate immunity (*Tan et al., 1999*). Pioneering studies demonstrated that evolutionarily conserved core signaling pathways including mitogen-activated protein kinase (MAPK) cascades (*Kim et al., 2002*), HLH-30/TFEB signaling (*Visvikis et al., 2014*) and β-catenin signaling (*Irazoqui et al., 2008*) regulate innate immunity from worms to mammals.

In *C. elegans*, the MAPK cascade of NSY-1-SEK-1-PMK-1 is essential to innate immunity, and has orthologs in mammals (ASK1 MAPKKK-MKK3/6 MAPKK-p38 MAPK pathway) playing a similar role (*Kim et al., 2002*). This signaling in *C. elegans* depends on TIR-1, a Toll-Interleukin-1 Receptor (TIR) domain protein and an ortholog of mammalian SARM (*Couillault et al., 2004*). Deletion mutants of the *C. elegans* PMK-1 pathway correspondingly exhibit dysregulated transcriptional responses and

hypersensitivity to infection challenge (*Fletcher et al., 2019*; *Kim et al., 2002*; *Troemel et al., 2006*). Evidently, pattern recognition receptors (PRRs), such as Toll-like-receptors, which function upstream to control the MAPK pathway in mammals and insects, do not appear to play a significant role in *C. elegans* infection responses (*Pujol et al., 2001*; *Pukkila-Worley and Ausubel, 2012*), but rather mediate the aversive behavioral response to pathogens (*Brandt and Ringstad, 2015*). G protein-coupled receptors (GPCRs) apparently play an important role in controlling the MAPK pathway in *C. elegans* in the context of infection (*Styer et al., 2008*; *Sun et al., 2011*; *Zugasti et al., 2014*). However, identities of the actual ligands during infection for these receptors are not clear except for *dcar-1*, whose ligand was identified to be an endogenous molecule, 4-hydroxyphenyllactic acid (HPLA) (*Zugasti et al., 2014*). Various stressors also trigger the MAPK pathway, but how these different inputs activate MAPK cascades in *C. elegans* is not well understood.

mRNA splicing is an essential process in eukaryotic cells whereby intervening non-coding sequences (introns) are removed from primary transcripts, and protein coding sequences (exons) are joined together to form the mature mRNA. This activity is catalyzed by a family of specialized proteins called splicing factors (*Wani and Kuroyanagi, 2017*). Different combinations of exons enlarge the repertoire of proteins and thus increase diversity of molecules at play in the physiological response. Various physiological processes are regulated by splicing in *C. elegans*, including heavy metal resistance, metabolism and lifespan (*Heintz et al., 2017*; *Tabrez et al., 2017*; *Wu et al., 2019a*), yet how splicing affects innate immunity, especially in invertebrate models, remains relatively unexplored. Here we report a novel role of splicing factors in immunity that reveal a crucial host response during bacterial infection.

## Results

### G281D substitution of RNP-6 enhances resistance to abiotic stresses

Cellular stress resistance mechanisms help manage diverse environmental and physiologic challenges. Animals have evolved adaptive mechanisms to survive heat and cold stress (*Morley and Morimoto, 2004*; *Robinson and Powell, 2016*). While heat stress mechanisms have been well studied, cold stress mechanisms are not well understood (*Ohta et al., 2014*). We observed that 2°C cold stress rapidly killed *C. elegans* wildtype N2 animals (*Figure 1A*), while long-lived strains such as *daf-2*/InsR and *eat-2* dietary restriction mutants were remarkably resistant (*Figure 1B*). In order to identify novel cold stress resistant loci, we performed an EMS screen for mutants able to withstand prolonged exposure (72 hr) to 2°C and recover to reproduce at normal growing temperature of 20°C (*Figure 1C*). Surviving mutants were outcrossed, and causative mutations were identified using genome sequencing and single-nucleotide polymorphism mapping (data not shown). Two screenings were conducted with 16000 haploid genomes screened. A total of 303 cold resistant mutants were recovered.

From the screen, we identified a novel mutation in *rnp-6*, which encodes an ortholog of the mammalian splicing factor Poly(U) Binding Splicing Factor 60 (PUF60). PUF60 facilitates the association of the U2 snRNP with target RNA molecules (*Page-McCaw et al., 1999*). PUF60 is essential, and heterozygous loss of function causes Verheij syndrome, which is characterized by severe developmental abnormalities in most patients (*Low et al., 2017*; *Verheij et al., 2009*). PUF60 is also involved in Hepatitis B virus infection (*Sun et al., 2017*) and cancer progression (*Sun et al., 2019*). In *C. elegans*, we found a G281D amino acid substitution in *rnp-6* lying within the second RNA binding motif at a position conserved across evolution (*Figure 1D*). Surprisingly, this mutation confers no obvious developmental phenotypes, despite *rnp-6* null mutants being embryonic lethal (*Johnsen et al., 2000*). Upon outcrossing the original mutant, *rnp-6(dh1124)* was highly resistant to cold stress (*Figure 1E*). An independent line carrying the same mutation (*dh1127*) was created using CRISPR gene editing, which showed the same phenotype, confirming the casual relation (*Figure 1E*). In subsequent experiments, we used the CRISPR line. The mutant had a slightly reduced brood size (*Figure 1—figure supplement 1A*), a slight but significant developmental delay (around 3.5 hr more to reach adulthood) (*Figure 1—figure supplement 1B*) and a marginally shorter body size (*Figure 1—figure supplement 1C*), but did not significantly affect pharyngeal pumping rate (*Figure 1—figure supplement 1D*). Interestingly, the mutant was also resistant to heat stress (*Figure 1F*) and oxidative stress (*Figure 1G*), suggesting this mutation widely confers resistance to multiple abiotic stresses.

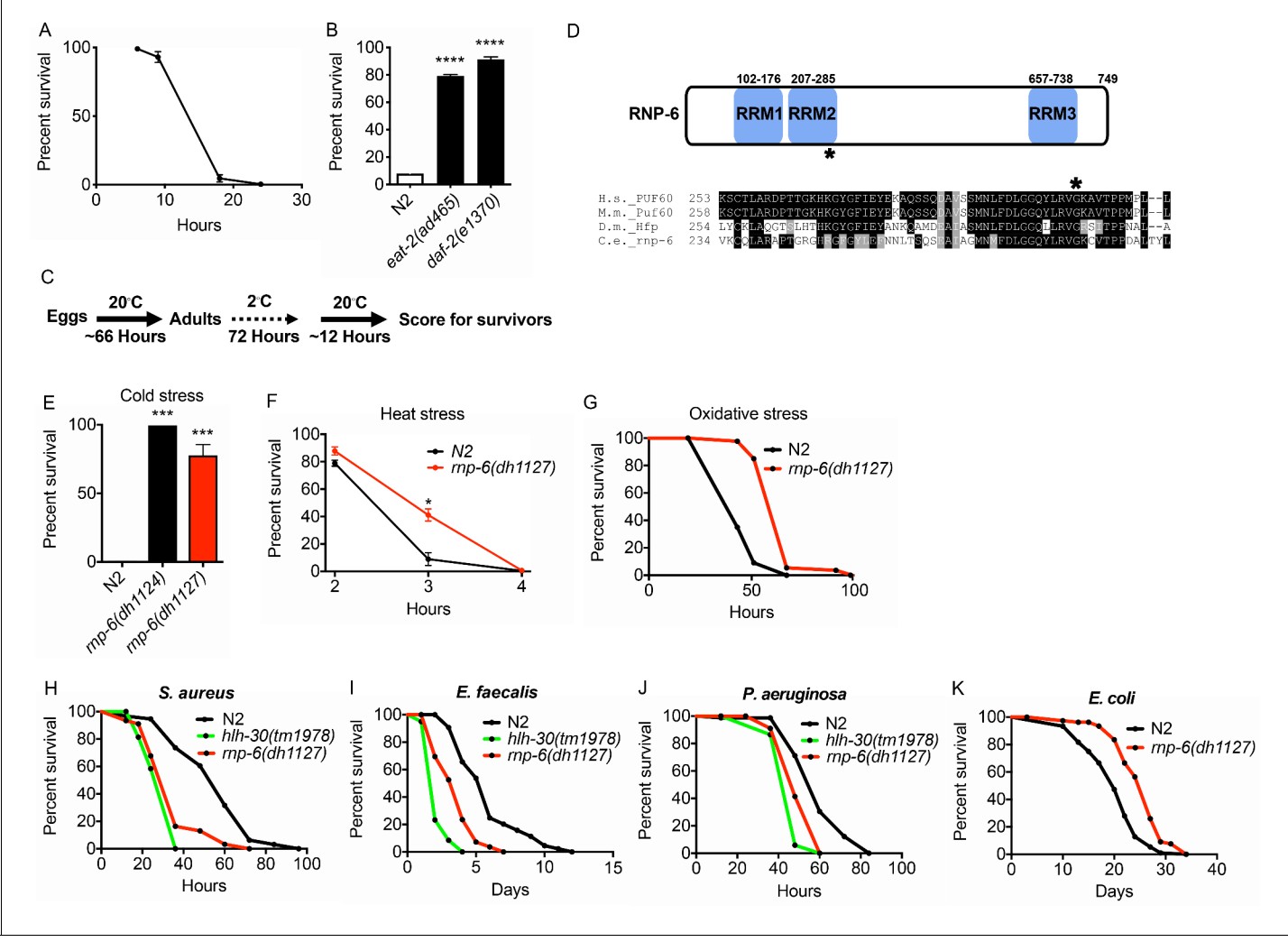

**Figure 1.** Isolation of the *rnp-6* G281D substitution mutant from a cold resistance screen. (**A**) Cold stress survival assay. Low temperature (2˚C) incubation kills wildtype young adult worms. (**B**) *daf-2* and *eat-2* mutants show enhanced survival after 24 hr incubation at 2˚C. (**C**) Schematic of the cold stress selection experiment. (**D**) Schematic showing the location of the G281D mutation in *rnp-6*. RRM stands for RNA recognition motif. The glycine at this position is conserved in human (*Homo sapiens*), mouse (*Mus musculus*) and fruit fly (*Drosophila melanogaster*). (**E**) The CRISPR strain (*dh1127*) shows the same cold resistant phenotype as the original mutant (*dh1124*) isolated from the genetic screen. (**F**) The *rnp-6* G281D mutant shows enhanced survival under 35˚C heat stress. (**G**) The *rnp-6* G281D mutant shows improved survival under oxidative stress (20 mM paraquat) (p<0.0001, log-rank test.). (**H,I,J**) Infection survival analysis. *rnp-6(dh1127)* animals show sensitivity to all the bacteria tested. *S. aureus* (p<0.0001, log-rank test.), *E. faecalis* (p<0.0001, log-rank test.) and *P. aeruginosa* (p=0.0022, log-rank test.). The *hlh-30* mutant serves as a positive control of infection sensitivity. (**K**) Demographic analysis of lifespan. *rnp-6(dh1127)* mutants have significant lifespan extension when cultured with OP50 bacteria at 20˚C (p<0.0001, log-rank test.). Survival and lifespan experiments were performed three times independently. Error bars represent mean ± s.e.m. *p<0.05, ***p<0.001, ****p<0.0001, unpaired t-test.

The online version of this article includes the following figure supplement(s) for figure 1:

**Figure supplement 1.** *rnp-6* G281D mutation compromises bacterial infection survival, related to *Figure 1*.

## G281D substitution of RNP-6 causes infection sensitivity

*C. elegans* mutants that are resistant to abiotic stresses often have better survival upon infectious insults and are long-lived (*Garsin et al., 2003*; *TeKippe and Aballay, 2010*). To test whether the stress resistant *rnp-6* G281D mutant also enhances infection survival, we infected the *rnp-6(dh1127)* mutant with different pathogenic bacteria, namely *Staphylococcus aureus*, *Enterococcus faecalis* and *Pseudomonas aeruginosa*. Unexpectedly, this mutant was more sensitive to all pathogenic bacteria tested (*Figure 1H–J*). This phenotype was confirmed with another independently created CRISPR line (*dh1125*) (*Figure 1—figure supplement 1E–G*). We also performed the experiments on full lawn

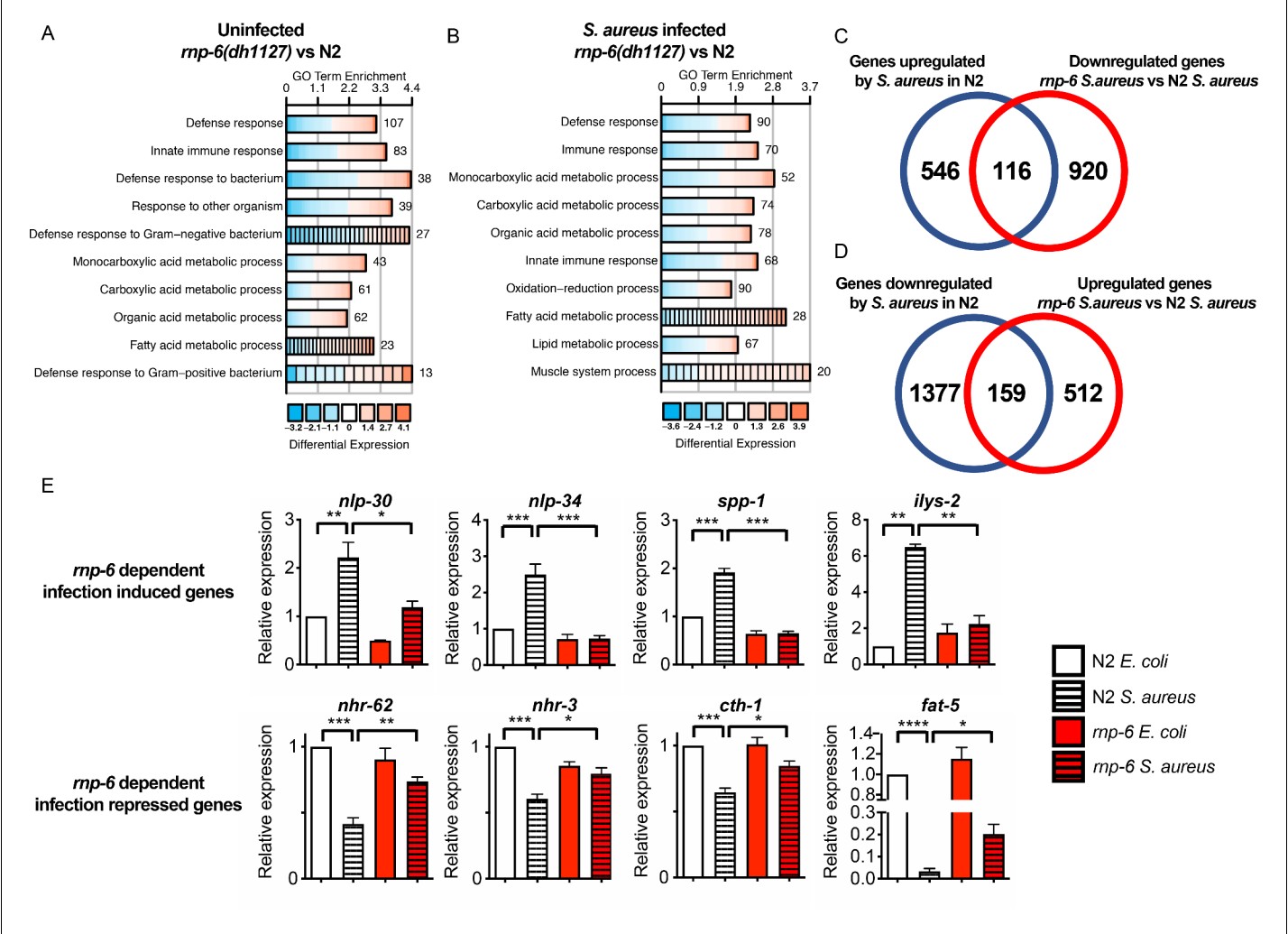

**Figure 2.** Dysregulation of immune responses in the *rnp-6* G281D mutant. (A,B) GO enrichment analysis using DAVID. RNA sequencing was performed on WT and *rnp-6(dh1127)* animals under control and *S. aureus* infected conditions. The 10 most statistically significant terms (based on p-values) are depicted for each comparison (p-values ascending). The enrichment (x axis), fold change (colour coding) and number of genes are shown. (C) Venn diagrams showing the numbers of up-regulated genes by infection and genes that are down-regulated in infected *rnp-6(dh1127)* when compared to infected N2 animals. *rnp-6* dependent infection induced gene are defined as the interception between the two set of genes. (D) Venn diagrams of genes that are down-regulated upon infection and genes that are up-regulated in infected *rnp-6(dh1127)* when compared to infected wildtype animals. *rnp-6* dependent infection repressed genes are defined as the interception between the two set of genes. (E) qRT-PCR results showing *rnp-6* G281D substitution compromises induction of antimicrobial genes (*nlp-30*, *nlp-34*, *spp-1* and *ilys-2*) upon *S. aureus* infection. Metabolic genes (*nhr-62*, *nhr-3*, *cth-1* and *fat-5*), which are suppressed in wildtype animals upon infection, maintain a higher expression in the *rnp-6(dh1127)* mutant. Error bars represent mean ± s.e.m. *p<0.05, **p<0.01, ***p<0.001, ****p<0.0001, unpaired t-test.

The online version of this article includes the following source data and figure supplement(s) for figure 2:

**Source data 1.** List of differentially expressed genes.
**Source data 2.** List of differentially expressed genes.
**Source data 3.** List of *rnp-6* dependent infection induced genes.
**Source data 4.** List of *rnp-6* dependent infection repressed genes.
**Figure supplement 1.** G281D substitution of *rnp-6* inhibits transcriptional responses to bacterial infection, related to *Figure 2*.

*S. aureus* plates. The *rnp-6(dh1127)* mutant displayed the same sensitive phenotype (*Figure 1—figure supplement 1H*), suggesting the sensitivity was not due to differential bacterial exposure. Although *rnp-6(dh1127)* animals were more sensitive to *S. aureus*, we did not observe a difference in bacterial burden (*Figure 1—figure supplement 1I*). Interestingly, when cultured with the weakly pathogenic *E. coli* strain OP50, the *rnp-6(dh1127)* mutant showed robust longevity at 20°C

(*Figure 1K*) and a normal lifespan at 25°C (*Figure 1—figure supplement 1J*), indicating the shortened survival on pathogenic bacteria was specific and unlikely due to general sickness of the mutant. Although longevity is dependent on temperature, the mutant was sensitive to pathogenic bacterial infection at both 20°C and 25°C (*Figure 1—figure supplement 1K*). Taken together, the data

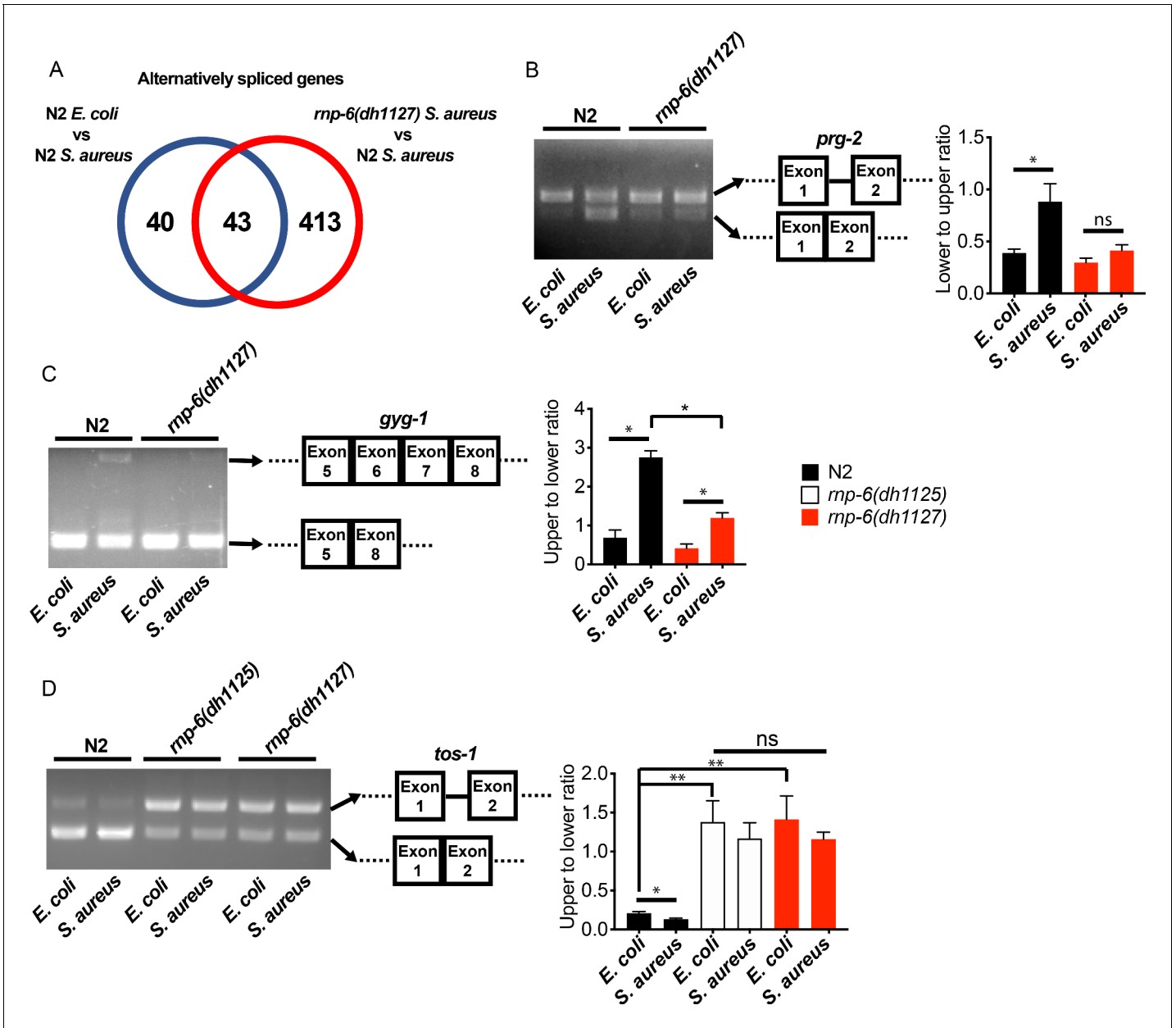

**Figure 3.** *rnp-6* remodels alternate splicing patterns upon infection. (**A**) Venn diagrams showing the numbers of alternatively spliced genes induced by *S. aureus* infection and those that are affected in infected *rnp-6(dh1127)* when compared to infected N2 animals. (**B,C,D**) Agarose gel analysis of RT-PCR products of *prg-2*, *gyg-1* and *tos-1*. In wildtype animals, *S. aureus* infection induces a shift in splicing patterns, which is suppressed in the *rnp-6* G281D mutants. Error bars represent mean ± s.e.m. *p<0.05, **p<0.01, ns non-significant, unpaired t-test.

The online version of this article includes the following source data and figure supplement(s) for figure 3:

**Source data 1.** List of alternatively spliced genes upon *S. aureus* infection in wildtype animals.

**Source data 2.** List of alternatively spliced genes between infected wildtype animals and infected *rnp-6(dh1127)* mutants.

**Source data 3.** List of genes whose splicing is affected by both infection and *rnp-6*.

**Figure supplement 1.** Alternative splicing events induced by infection and *rnp-6* G281D substitution, related to *Figure 3*.

suggest G281D substitution of *rnp-6* enhances abiotic stress resistance, prolongs lifespan, but specifically compromises infection survival.

Next, we assessed the levels of RNP-6 upon infection. Using CRISPR technology, we tagged endogenous RNP-6 with a N-terminal HA tag (*dh1188*). We performed qRT-PCR and western blots to detect the endogenous levels of RNP-6 at both transcript and protein levels. However, no significant changes could be detected after *S. aureus* infection (*Figure 1—figure supplement 1L,M*). We conclude that although RNP-6 profoundly affects infection survival, its levels remain stable upon infection in *C. elegans*.

## G281D substitution of RNP-6 leads to dysregulation of immune responses

To further investigate the effect of RNP-6 on infection responses, we performed transcriptomic profiling of wildtype N2 animals and the *rnp-6* G281D mutant, under both control *E. coli* OP50 and *S. aureus* infected conditions. The *rnp-6* G281D mutation led to numerous differentially expressed genes (DEGs) under both non-infected (1483 genes) (*Figure 2—source data 1*) and infected conditions (1707 genes) (*Figure 2—source data 2*). Under both conditions, Gene Ontology (GO) term analysis of the DEGs revealed an enrichment of genes implicated in immunity (*Figure 2A,B*). 662 genes were found to be induced by *S. aureus* infection in the N2 background. For 116 of them, the expression was dampened in the *rnp-6* mutant under infection (*Figure 2C*) (*Figure 2—source data 3*), and these *rnp-6* dependent infection induced genes were enriched for immune response genes (*Figure 2—figure supplement 1A*). Conversely, 1536 genes were repressed by *S. aureus* infection, and the expression remained higher in the *rnp-6* mutant for 159 of them (*Figure 2D*) (*Figure 2—source data 4*). Among these *rnp-6* dependent infection repressed genes, metabolic genes appeared to be highly enriched (*Figure 2—figure supplement 1B*). The most enriched GO terms included organic acid metabolic process and fatty acid metabolic process (*Figure 2—figure supplement 1B*).

qRT-PCR was performed on selected genes. Consistent with the results from RNA sequencing, infection induced expression of antimicrobial proteins (*nlp-30*, *nlp-34* and *spp-1*) and lysozyme (*ilys-2*) was severely compromised by *rnp-6* G281D substitution (*Figure 2E*). Other categories of infection induced genes were also affected by *rnp-6* (*Figure 2—figure supplement 1C*). On the other hand, the expression of nuclear hormone receptors (*nhr-62* and *nhr-3*) and metabolic enzymes (*cth-1* and *fat-5*) was refractory to the repression by infection in the *rnp-6* G281D mutant (*Figure 2E*). All in all, we conclude that G281D substitution of RNP-6 causes dysregulation of immune functions and inhibits transcriptional responses to bacterial infection.

## RNP-6 regulates splicing effects of bacterial infection

We next wondered whether splicing was affected by the *rnp-6* G281D mutation. We analyzed splicing patterns of the transcriptomic data using Cufflinks (*Trapnell et al., 2012*) and KISSDE (*Lopez-Maestre et al., 2016*) protocols. 83 genes (*Figure 3A*, *Figure 3—source data 1*) were found to be alternatively spliced upon *S. aureus* infection in wildtype animals. When comparing infected wildtype animals and infected *rnp-6(dh1127)* mutants, we found a larger number of alternatively spliced genes (456 genes) (*Figure 3A*, *Figure 3—source data 2*). The number of alternatively spliced genes induced by *S. aureus* could be underestimated as a result of the short infection time used (4 hr post-infection) and tissue-specific alternative splicing masked in the transcriptome from whole animals. Over half of the genes (43 out of 83) (*Figure 3A*, *Figure 3—source data 3*) showing infection induced alternative splicing, were also altered in the infected *rnp-6(dh1127)* mutants when compared to infected wildtype N2 animals. This suggests that the splicing alteration induced by infection is partly dependent on *rnp-6*. We also analyzed the data with the SAJR algorithm, which gives a detailed classification of the splicing events (*Mazin et al., 2013*). In wildtype animals, *S. aureus* infection induced splicing events were predominated by 'cassette exon' (44.9%) and 'retained intron' (22.4%). This was also true when we compared infected *rnp-6(1127)* mutants with infected wildtype animals ('cassette exon' 59.5%, 'retained intron' 15.2%) (*Figure 3—figure supplement 1A,B*). Using RT-PCR and agarose electrophoresis, we analyzed the splicing patterns of *prg-2*, *gyg-1* and *tos-1*. whose splicing is both infection and *rnp-6* dependent. *S. aureus* infection enhanced intron one splicing in *prg-2* in wildtype animals, but this effect was substantially suppressed in the *rnp-6(dh1127)*

mutants (*Figure 3B*). Infection also led to inclusion of exon 6 and exon 7 of *gyg-1*, and again this effect was largely suppressed by the G281D substitution in the *dh1127* mutants (*Figure 3C*). Further, infection increased the abundance of the short *tos-1* isoform in wildtype animals (*Figure 3D*). Intriguingly, the *rnp-6* G281D mutants had an increased longer to shorter isoform ratio under non-infected condition, and this ratio remained the same after infection (*Figure 3D*), suggesting it is refractory to input. Taken together, these data show that bacterial infection induces an RNP-6 dependent splicing remodeling in *C. elegans*.

## The activity of RNP-6 impairs immunity

The G281D substitution of *rnp-6* compromises immunity (*Figures 1* and *2*), but how this substitution affects RNP-6 activity still remains elusive. We next sought to investigate the link between the activity levels of RNP-6 and innate immunity. Surprisingly, RNAi knockdown of *rnp-6* resulted in a robust extension of survival upon *S. aureus* infection (*Figure 4A*). Knock-down of many essential genes is reported to induce microbial aversion (*Melo and Ruvkun, 2012*). To test whether the survival phenotype of *rnp-6* RNAi is due to reduced exposure to bacteria, we repeated the survival experiment using full lawn plates on which bacterial avoidance is impossible. RNAi against *rnp-6* still conferred enhanced survival under these conditions (*Figure 4—figure supplement 1A*), ruling out the possibility that the phenotype is caused by differential bacterial exposure. Further, *rnp-6* RNAi did not affect *S. aureus* bacterial burden, suggesting the survival phenotype is caused by independent mechanisms (*Figure 4—figure supplement 1B*).

Consistent with the survival results, *rnp-6* RNAi strongly induced the expression of numerous infection responsive genes, including *nlp-34*, *lys-3*, *irg-2*, *irg-1*, *M01G12.9*, *fmo-2* and *cyp-37B1* (*Figure 4B*). These findings suggest that RNP-6 normally suppresses immunity, and that reduction of function by RNAi derepresses immunity. Notably, among the genes that are transcriptionally induced by *rnp-6* RNAi, several of the same genes, namely *nlp-34*, *M01G12.9* and *cyp-37B1*, were suppressed by the *rnp-6* G281D substitution under *S. aureus* infection (*Figure 2E*, *Figure 2—figure supplement 1C*). The G281D substitution mutation, in this regard, behaved oppositely to *rnp-6* RNAi (marked with A in *Figure 4B*). In addition, other genes highly induced by *rnp-6* RNAi, such as *lys-3* and *fmo-2* (*Figure 4B*), were unaffected by the *rnp-6* G281D substitution under infection conditions (*Figure 4—figure supplement 1C*). Moreover, the levels of *lys-3*, *irg-2*, *irg-1* and *fmo-2* (marked with B in *Figure 4B*), which were induced upon *rnp-6* RNAi, were also slightly elevated in the *rnp-6* G281D mutants under non-infected condition (*Figure 4—figure supplement 1C*). These results indicate that *rnp-6* G281D substitution and *rnp-6* RNAi resulted in partially overlapping and partially opposite transcriptional responses of immune genes.

Next, we assessed how *rnp-6* RNAi affects splicing by measuring *prg-2, tos-1* and *gyg-1* transcripts as above. Splicing of *prg-2* and *tos-1*, which exhibited *rnp-6* G281D dependent splicing remodeling upon infection (*Figure 3*), was also affected by *rnp-6* RNAi, whereas *gyg-1* had no detectable changes (*Figure 4—figure supplement 1D*). Notably, *rnp-6* RNAi promoted the splicing of *prg-2* intron-1 (*Figure 4C*), which shows the same trend as seen during infection and is largely inhibited by the G281D mutation (*Figure 3B*). On the other hand, *rnp-6* RNAi increased the relative abundance of the longer isoform of *tos-1* (*Figure 4D*), which more resembles the change induced by the G281D mutation (*Figure 3D*). These results confirm that *rnp-6* modulates the splicing of infection induced alternatively spliced genes and indicate that *rnp-6* G281D substitution and *rnp-6* RNAi resulted in partially overlapping and partially different transcriptional and splicing responses (See Discussion). Furthermore, overexpressing RNP-6 greatly compromised infection survival upon *S. aureus* (*Figure 4E*) and *E. faecalis* exposure (*Figure 4—figure supplement 1E,F*), but promoted longevity at 25°C when the animals were cultured with the weakly pathogenic *E. coli* OP50 (*Figure 4F*). The sensitivity phenotype of the RNP-6 overexpressing strain is not due to differential bacterial exposure, since the phenotype was still manifest on full lawns (*Figure 4—figure supplement 1G*). Also, overexpression of RNP-6 did not have an observable effect on bacterial burden in *S. aureus* infected *C. elegans* (*Figure 4—figure supplement 1H*). Taken together, these results suggest that the activity of RNP-6 is detrimental to immunity and survival upon infection, but promotes longevity under non-infected conditions. Like the overexpressor, the G281D substitution is also immuno-suppressive and lifespan-extending, indicating that the substitution resembles a gain-of-function mutation in this context (see Discussion).

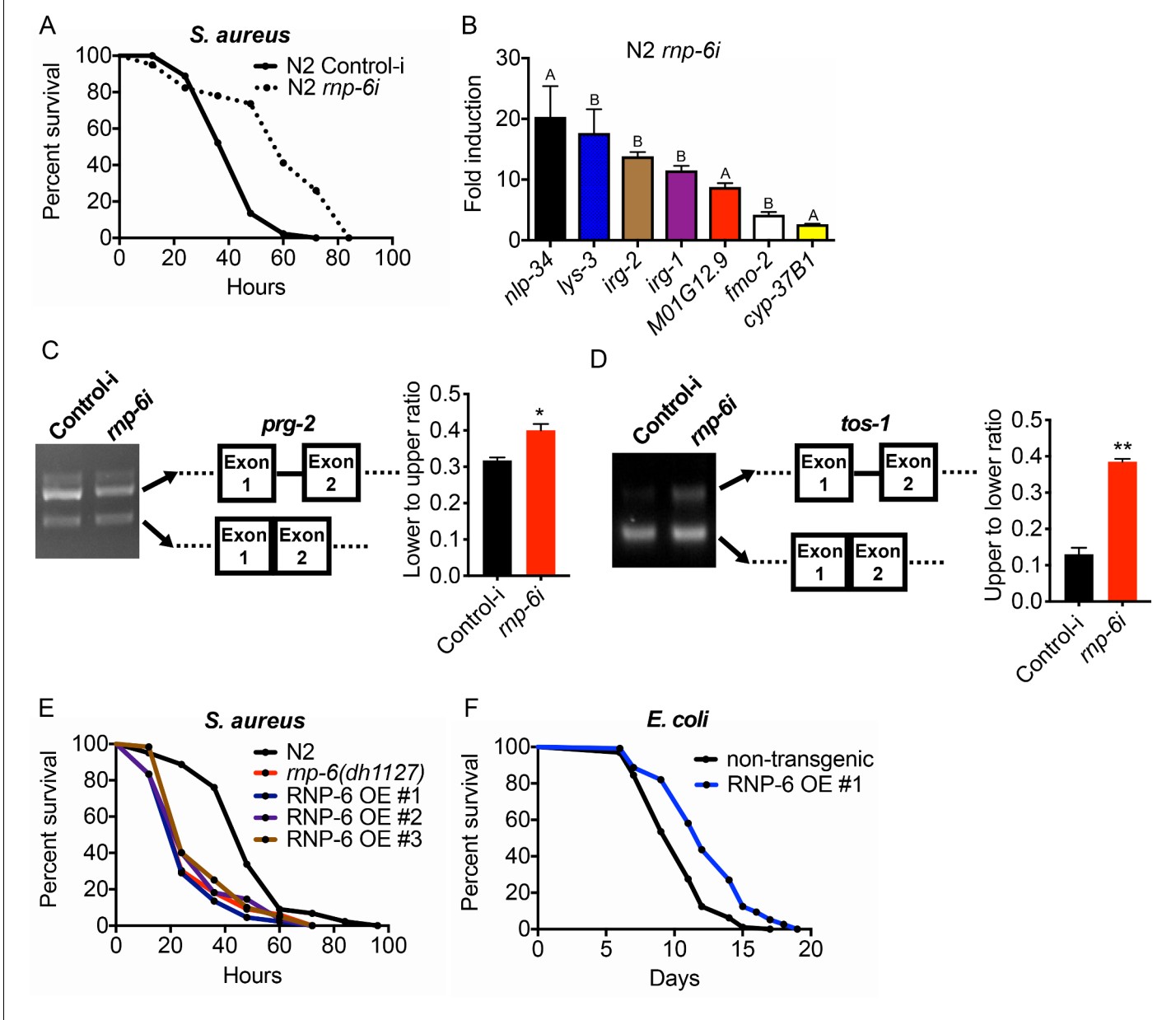

**Figure 4.** RNP-6 restrains innate immunity in *C. elegans*. (**A**) Infection survival analysis. Young adult worms fed with either control or *rnp-6* RNAi HT115 bacteria were infected with *S. aureus. rnp-6* RNAi leads to prolonged survival (p<0.0001, log-rank test.). (**B**) qRT-PCR results showing that wildtype animals grown on *rnp-6* RNAi HT115 bacteria have elevated levels of immune-related genes. (*nlp-34* p=0.0184, *lys-3* p=0.0132, *irg-2* p<0.0001, *irg-1* p=0.0002, *M01G12.9* p=0.0002, *fmo-2* p=0.0021, *cyp-37B1* p<0.0001, unpaired t-test.). 'A' denotes genes that are repressed in the *rnp-6* G281D mutant under infection condition. 'B' denotes genes that are also slightly induced in the *rnp-6* G281D mutant. Error bars represent mean ± s.e.m. (**C,D**) Agarose gel analysis of RT-PCR products of *prg-2* and *tos-1*. In wildtype animals, *rnp-6* RNAi (OP50) induces a shift in splicing patterns. (**E**) RNP-6 overexpressing strains show enhanced sensitivity upon infection with *S. aureus* (RNP-6 OE #1, p<0.0001, RNP-6 OE #2 P<0.0001, RNP-6 OE #3 p=0.0006, log-rank test). (**F**) Demographic analysis of lifespan. Overexpression of RNP-6 prolongs lifespan when the animals are cultured with *E. coli* OP50 at 25°C (p<0.0001, log-rank test.). Survival and lifespan experiments were performed three times independently. Error bars represent mean ± s.e.m. *p<0.05, **p<0.01, unpaired t-test.

The online version of this article includes the following figure supplement(s) for figure 4:

**Figure supplement 1.** *rnp-6* regulates immune gene expression, and its overexpression compromises infection survival, related to *Figure 4*.

## RNP-6 inhibits PMK-1 MAPK signaling

We next sought to investigate mechanisms downstream of RNP-6. We used an online bioinformatic tool (http://wormexp.zoologie.uni-kiel.de/wormexp/) (*Yang et al., 2016*) to investigate the similarity of the transcriptomic data of infection responses of the G281D mutant of *rnp-6*. Interestingly, we found that the *rnp-6* dependent infection responsive genes were significantly enriched for PMK-1 target genes (*Figure 5A*). Among the 116 RNP-6 dependent infection induced genes (*Figure 2C*), 17 were also PMK-1 dependent (*Bond et al., 2014*; *Figure 5—source data 1*). For the 159 genes whose repression by infection was reverted by the *rnp-6* G281D substitution (*Figure 2D*), 55 of them were also up-regulated in the *pmk-1* deletion mutant (*Mertenskötter et al., 2013*; *Figure 5— source data 2*). These results led us to hypothesize an involvement of PMK-1.

To test this hypothesis, we examined the genetic epistasis between *rnp-6* and *pmk-1*. Knocking down *rnp-6* increased the animals' survival upon *S. aureus* infection, and deletion of *pmk-1* or its upstream activator *tir-1* completely abolished the beneficial effect of *rnp-6* reduction (*Figure 5B,C*). By contrast, deletions of *daf-16*/FOXO or *hlh-30*/TFEB, which are also important regulators of innate immunity (*Garsin et al., 2003*; *Visvikis et al., 2014*), had no and minimal effects on the *rnp-6* RNAi extended survival, respectively (53.6% increase in N2% and 43.4% increase in *hlh30(tm1978)*, area under survival curves) (*Figure 5—figure supplement 1A,B*). Overexpressing RNP-6 or G281D substitution rendered the animals susceptible to *S. aureus* infection in wildtype background, but did not further enhance sensitivity in the *pmk-1* deletion background (*Figure 5D,E*). The *rnp-6* G281D substitution also did not further sensitize the *pmk-1* deletion mutant under *P. aeruginosa* infection conditions (*Figure 5F*). We conclude that the sensitivity caused by *rnp-6* G281D substitution or RNP-6 overexpression and *pmk-1* deletion are not additive, suggesting the two genes function in the same pathway to modulate infection survival.

PMK-1 becomes phosphorylated upon activation. Given the observed genetic interactions between *pmk-1* and *rnp-6*, we next sought to test whether RNP-6 affects the activity of PMK-1 by measuring phospho-PMK-1. Upon RNAi knockdown of *rnp-6*, we observed a significant increase in phospho-PMK-1 (*Figure 5G*), suggesting PMK-1 activity increases upon *rnp-6* reduction. This is consistent with the immuno-suppressive role of *rnp-6*. As our results imply that RNP-6 modulates PMK-1 MAPK signaling, we asked whether the splicing patterns of the genes involved in this pathway were affected by the *rnp-6* G281D mutation. However, we did not observe any reproducible difference in splicing patterns or expression levels of PMK-1 MAPK signaling components comparing wildtype and *rnp-6(dh1127)* animals under non-infected or *S. aureus* infected conditions (*Figure 5—figure supplement 2A–I*; *Figure 2—source data 1* and *2*), suggesting that *rnp-6* regulates this pathway via other mechanisms. Altogether, both genetic and biochemical analyses imply that RNP-6 exerts its inhibitory effects on immunity by suppressing the activity of PMK-1.

## SFA-1 acts downstream of RNP-6 to restrain immunity and promote longevity

Splicing has recently emerged as a determinant of animal lifespan (*Heintz et al., 2017*; *Tabrez et al., 2017*). Since *rnp-6* promotes longevity (*Figures 1K* and *4F*) but compromises immunity (*Figure 4*), we suspected other lifespan promoting splicing factors may also inhibit immunity. Overexpressing splicing factor SFA-1 resulted in robust longevity under non-infected conditions (*Figure 6A*), as previously reported (*Heintz et al., 2017*). However, similar to *rnp-6*, SFA-1 overexpression severely compromised survival upon *S. aureus* (*Figure 6B*) and *E. faecalis* infection (*Figure 6—figure supplement 1A*). Moreover, RNAi knockdown of *sfa-1* not only increased infection survival in wildtype worms (*Figure 6C*), but also completely suppressed infection sensitivity of the G281D mutant of *rnp-6* (*Figure 6D*). Consistent with a strong interaction, SFA-1 overexpression did not further increase *rnp-6* G281D longevity when animals were cultured with *E. coli* OP50 (*Figure 6E*), and knocking down *sfa-1* completely suppressed *rnp-6* G281D longevity (*Figure 6F*). By contrast, knocking-down *repo-1*, which is also implicated in longevity (*Heintz et al., 2017*), only partially rescued the infection sensitivity caused by *rnp-6* G281D mutation (*Figure 6—figure supplement 1B*), while *hrpu-1* knockdown had little effect (*Figure 6—figure supplement 1C*).

Given the strong genetic interaction with SFA-1, we tested whether RNP-6 and SFA-1 physically interact in *C. elegans*. However, we saw no obvious co-immunoprecipitation of RNP-6 with SFA-1 (*Figure 6—figure supplement 1D*). Thus, these factors probably function in structurally or

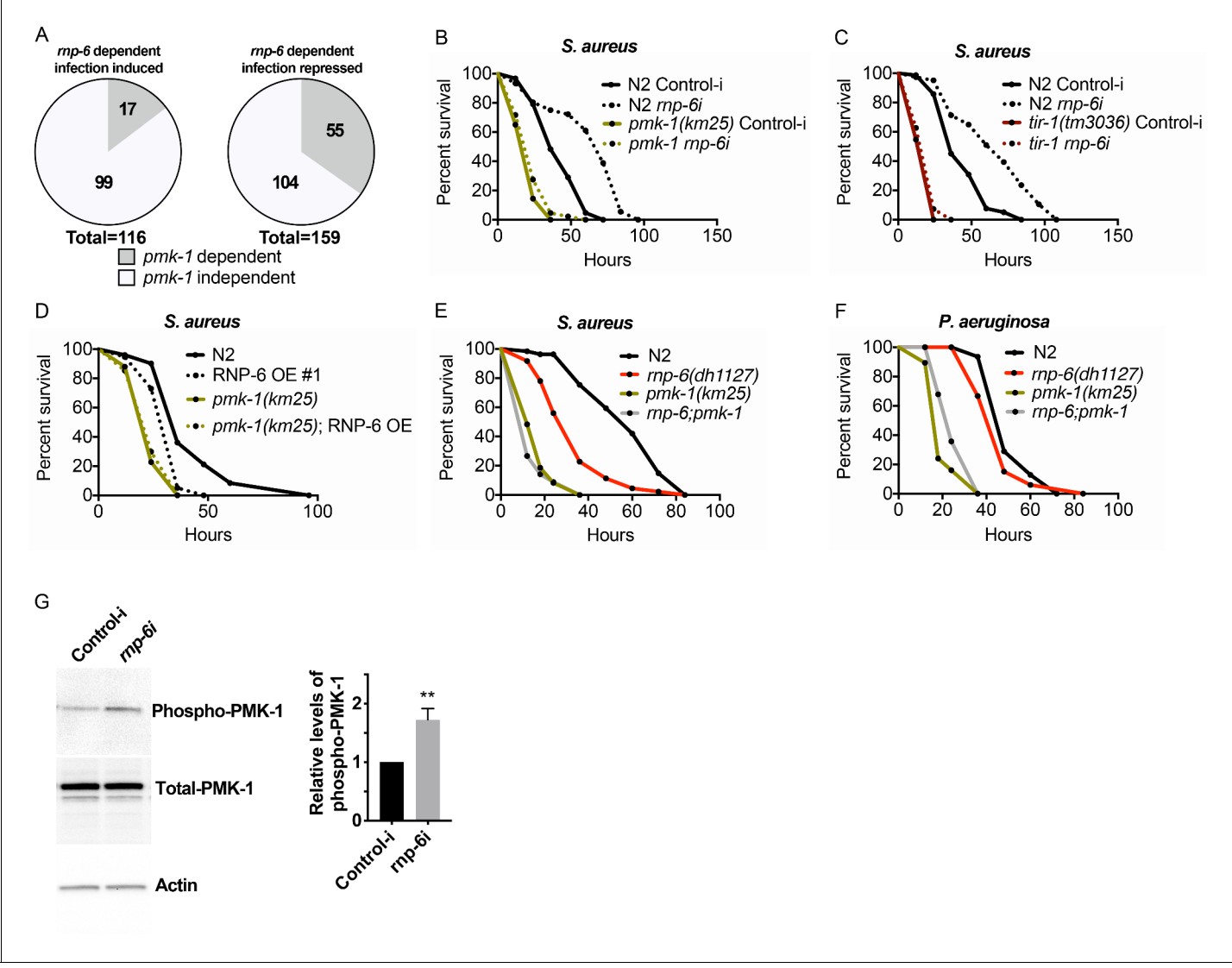

**Figure 5.** RNP-6 controls PMK-1 activity. (**A**) Pie charts showing the portion of *rnp-6* dependent infection responsive genes which are also reported to be *pmk-1* dependent. 17 out of 116 *rnp-6* dependent infection induced genes also have *pmk-1* dependent induction (p<0.0001). 55 out of 159 *rnp-6* dependent infection repressed genes are up-regulated in *pmk-1* deletion mutants (p<0.0001). (**B,C**) Infection survival analysis. Wildtype, *pmk-1(km25)* and *tir-1(tm3036)* animals were fed with either control or *rnp-6* RNAi OP50 bacteria and then infected with *S. aureus*. *rnp-6* RNAi does not prolong survival in the *pmk-1* or *tir-1* deletion mutant (non-significant, log-rank test.). (**D**) Overexpression of RNP-6 does not further sensitize *pmk-1* deletion mutants to *S. aureus* infection (non-significant, log-rank test.). (**E**) *rnp-6(dh1127)* does not further sensitize the *pmk-1* deletion mutants to *S. aureus* infection (non-significant, log-rank test.). (**F**) *rnp-6(dh1127)* mutation decreases survival in wildtype background upon *P. aeruginosa* infection (p=0.0056, log-rank test). However, the double mutant of *rnp-6(dh1127)* and *pmk-1(km25)* has enhanced survival relative to *pmk-1(km25)* alone (p<0.0001, log-rank test). (**G**) Western blot analysis showing the levels of phosphorylated PMK-1. N2 animals grown on HT115 RNAi bacteria targeting *rnp-6* possess higher levels of PMK-1 phosphorylation. **p<0.05. Error bars represent mean ± s.e.m., unpaired t-test. Survival experiments were performed three times independently.

The online version of this article includes the following source data and figure supplement(s) for figure 5:

**Source data 1.** List of *rnp-6* dependent infection induced genes with *pmk-1* dependency.
**Source data 2.** List of *rnp-6* dependent infection repressed genes with *pmk-1* dependency.
**Figure supplement 1.** *daf-16* and *hlh-30* do not play a significant role in *rnp-6* mediated immunity, related to *Figure 5*.
**Figure supplement 2.** *rnp-6* regulates immunity independently of splicing of the PMK-1 MAPK signaling pathway genes, related to *Figure 5*.

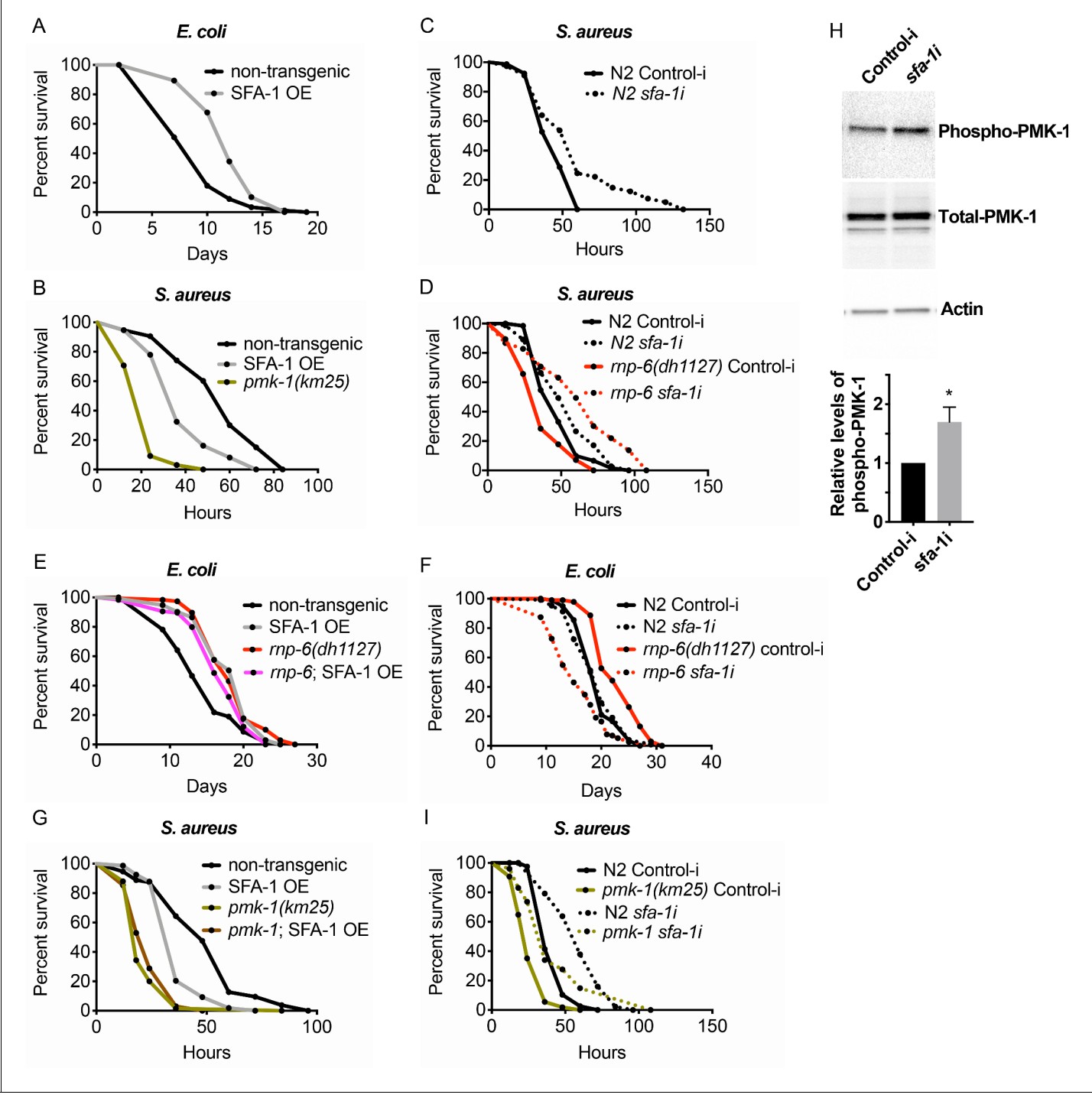

**Figure 6.** Suppression of immunity by SFA-1. (**A**) Demographic analysis of lifespan. Overexpression of SFA-1 extends lifespan when the animals are cultured with *E. coli* OP50 at 25°C (p<0.0001, log-rank test.). (**B**) Infection survival analysis. SFA-1 overexpressing animals show significant sensitivity to *S. aureus* infection (p=0.0165, log-rank test.). The *pmk-1* deletion mutant serves as a positive control for infection sensitivity. (**C**) Wildtype worms grown on control or *sfa-1* RNAi OP50 bacteria were infected with *S. aureus*. *sfa-1* knockdown improves survival (p=0.0048, log-rank test.). (**D**) *sfa-1* knockdown using OP50 RNAi bacteria suppresses infection sensitivity of *rnp-6(dh1127)* mutants (p<0.0001, log-rank test, comparing *rnp-6(dh1127)* Control-i and *rnp-6 sfa-1i*). (**E**) Demographic analysis of lifespan. SFA-1 overexpression does not further extend the lifespan of *rnp-6(dh1127)* mutants. In fact, SFA-1 overexpression slightly reduces the *rnp-6* mutants' lifespan (p=0.0147, log-rank test.). (**F**) *sfa-1* RNAi (OP50) abolishes *rnp-6(dh1127)* longevity (p<0.0001, log-rank test, comparing *rnp-6(dh1127)* Control-i and *rnp-6 sfa-1i*). (**G**) SFA-1 overexpression does not further sensitize *pmk-1(km25)* animals to *S. aureus* infection (non-significant, log-rank test.). (**H**) Western blot analysis showing the levels of phosphorylated PMK-1. N2 animals grown on OP50 RNAi bacteria targeting *sfa-1* possess higher levels of PMK-1 phosphorylation. Error bars represent mean ± s.e.m., *p<0.05, unpaired t-test. (**I**) Infection

*Figure 6 continued*

survival analysis. *sfa-1* knockdown using OP50 RNAi bacteria extends infection survival upon *S. aureus* infection in *pmk-1(km25)* mutants (p<0.0001, log-rank test.). Experiments of panel A and E were performed twice. All other survival and lifespan experiments were performed three times independently. The online version of this article includes the following figure supplement(s) for figure 6:

**Figure supplement 1.** *sfa-1* specifically work with *rnp-6* to regulate infection survival, related to *Figure 6*.

temporally distinct steps during the splicing process. We also observed that the sensitivity caused by SFA-1 overexpression was not additive to that of *pmk-1* deletion (*Figure 6G*), supporting the idea of *sfa-1* and *pmk-1* working in the same pathway, similar to that of *rnp-6*. Next, we tested whether *sfa-1* also exerts inhibitory effects on PMK-1 activity. Knocking-down *sfa-1* also induced phosphorylation of PMK-1 (*Figure 6H*), indicating that *sfa-1* similarly restrains PMK-1 activation. In contrast to *rnp-6* RNAi, however, *sfa-1* RNAi was able to increase infection survival in *pmk-1* deletion mutants (*Figure 6I*), suggesting *sfa-1* also moderates immunity via *pmk-1* independent pathways. Taken together, these results point towards a similar suppression of immunity by both RNP-6 and SFA-1 through their modulation of the PMK-1 MAPK signaling pathway, with SFA-1 working downstream of RNP-6.

## Mammalian PUF60 exerts immuno-suppressive and anti-inflammatory effects

Because PUF60 is involved in various human diseases (*Sun et al., 2019*; *Sun et al., 2017*; *Xu et al., 2018*), we next explored the potential role of *rnp-6* in imparting immunity in higher organisms. We first asked if PUF60, the mammalian homolog of *rnp-6*, is regulated during infection. We infected human epithelial HeLa cells, mouse embryonic fibroblasts (MEF) and RAW264.7 macrophage-like cells with *S. aureus* and observed a sharp reduction of PUF60 protein levels in all three cell types (*Figure 7A*). *S. aureus* infection also reduced PUF60 transcript levels (*Figure 7—figure supplement 1A*). Therefore, mammalian PUF60 responds to infection by reducing its expression, while *C. elegans rnp-6* levels remain stable after infection (*Figure 1—figure supplement 1L,M*). Next, we wondered whether this reduction of PUF60 has an impact on the immune-response and utilized siRNA to reduce PUF60 expression in HeLa cells (*Figure 7—figure supplement 1B*). Interestingly, reduction of PUF60 alone was sufficient to drive expression of pro-inflammatory cytokines, namely IL-6, IL-8, CXCL2, IL-1α (*Figure 7B*), IL-18 and CCL5 (*Figure 7—figure supplement 1C*), but had no effects on anti-inflammatory gene TGF-β1 (*Figure 7—figure supplement 1D*). We also observed a moderate induction of the same cytokines upon PUF60 knockdown in RAW264.7 macrophage-like cells (*Figure 7—figure supplement 1E*) despite the relatively low RNAi efficiency of PUF60 in this cell line (*Figure 7—figure supplement 1B*). Similar results were observed in *C. elegans*, where RNAi against *rnp-6* also induced infection responsive genes (*Figure 4B*), indicating an evolutionarily conserved role of RNP-6/PUF60 of suppressing basal immune-gene expression.

To further investigate the effects of PUF60 on cytokine expression, we constructed the mammalian version of the *C. elegans* gain of function G281D substitution. This glycine residue is conserved across evolution (*Figure 1D*) and corresponds to a G to D substitution at the 300[th] amino acid in human PUF60. Equal amounts of the expression plasmids of wildtype and the mutated PUF60 were transfected into HeLa cells. The protein levels of the mutated PUF60 were lower compared to the wildtype form (*Figure 7—figure supplement 1G*), even though the expression levels of mRNA were similar (*Figure 7—figure supplement 1F*), suggesting the mutation may destabilize the protein. Intriguingly, overexpression of the mutated form of PUF60 stimulated expression of anti-inflammatory TGF-β1, while the wildtype PUF60 had no effects on TGF-β1 (*Figure 7—figure supplement 1D*). Furthermore, overexpression of the mutated PUF60 significantly reduced inflammation induced by *S. aureus* infection in HeLa cells. The induction of IL-8, IL-6, CXCL2 and IL-1α upon infection was significantly suppressed by overexpressing the mutated PUF60 (*Figure 7C*). Overexpression of wildtype PUF60 also exerted moderate inhibitory effects on cytokine induction but the effects were not statistically significant (*Figure 7C*). Also, overexpression of wildtype or the mutated form of PUF60 improved HeLa cell survival upon *S. aureus* infection at a later time point (post-infection 24 hr) (*Figure 7D*) without affecting bacterial internalization (*Figure 7—figure supplement 1H*). Enhanced survival might arise due to reduced inflammation mediated by PUF60. Remarkably then, the G281D

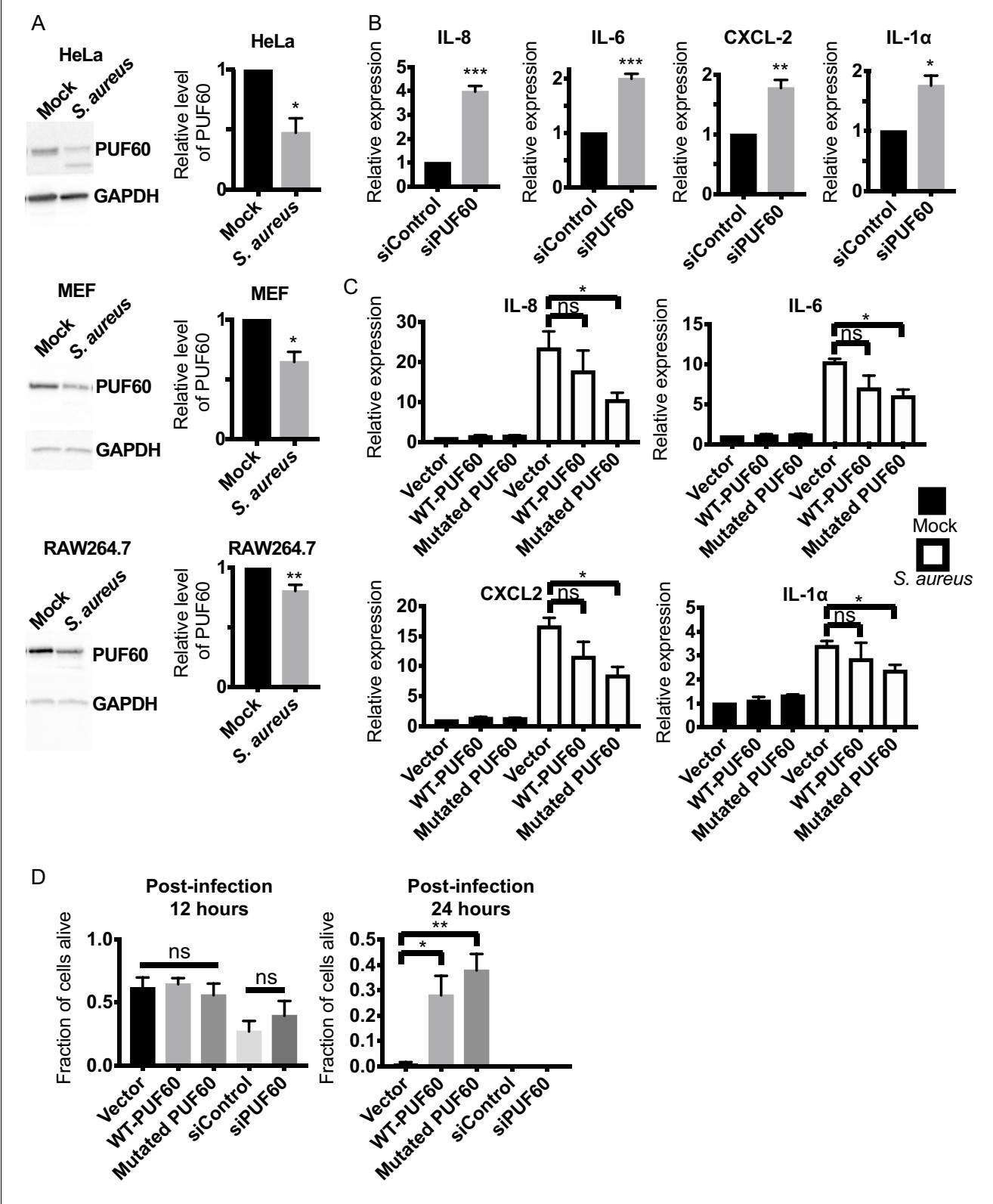

**Figure 7.** Evolutionary conservation of immuno-suppressive and anti-inflammatory functions of PUF60 in mammalian cells. (**A**) Western blot analysis of *S. aureus* infected HeLa, MEF and RAW264.7 cells at 6 hr post-infection. *S. aureus* infection significantly reduces the protein levels of PUF60. (**B**) qRT-PCR results. HeLa cells were transfected with either control non-targeting or PUF60 siRNA. The cells were harvested at post-transfection 48 hr. PUF60 RNAi induces cytokine expression. GAPDH was used for internal normalization. (**C**) qRT-PCR results of *S. aureus* infected HeLa cells. Cells were

*Figure 7 continued on next page*

*Figure 7 continued*

transfected with the indicated expression plasmids. At 48 hr post-transfection, the cells were infected with *S. aureus* and harvested at 6 hr post-infection. Transfection of the mutated PUF60 (G300D) plasmid significantly reduces the induction of pro-inflammatory cytokines. Although the trend of cytokine suppression is evident for wildtype PUF60, the effects do not reach statistical significance. RPL13A was used for internal normalization. (D) HeLa cells survival after *S. aureus* infection. Overexpression of either wildtype or the mutated PUF60 (G300D) improves cell survival at 24 hr post-infection. Cells were transfected with the indicated siRNA or expression plasmids and then infected with *S. aureus* 48 hr after transfection. Error bars represent mean ± s.e.m. *p<0.05, **p<0.01, ***p<0.001, ns non-significant, unpaired t-test.

The online version of this article includes the following figure supplement(s) for figure 7:

**Figure supplement 1.** PUF60 suppresses inflammation in mammalian cells, related to *Figure 7*.

mutation, originally identified in *C. elegans*, also works to enhance the immune-suppression of RNP-6/PUF60 in mammalian cells, and the function of RNP-6/PUF60 in suppressing immunity is highly conserved in evolution.

## Discussion

Our data reveal that the splicing factor RNP-6/PUF60 is a novel regulator of immunity and lifespan. Viruses commonly interact with the host splicing machinery, whereas not much is known regarding bacterial interference with host RNA splicing (*Chauhan et al., 2019*). We provide evidence that in *C. elegans*, bacterial infections affect mRNA splicing, and perturbations of splicing is a trigger for immune responses while, by inference, increased splicing activity compromises immunity, thus providing a new explanation for how immune responses are modulated in the worms. Importantly, the function of RNP-6/PUF60 in immunity is conserved in mammalian cells. Thus, our results also shed light on the mechanisms of human diseases that involve PUF60.

Long-lived animals are often more stress resistant. Stress resistance and longevity are thought to be tightly and positively correlated, but this idea has been recently disputed (*Amrit et al., 2019*). The behavior of splicing factors RNP-6 and SFA-1 is another clear example of disconnecting longevity and biotic stress resistance. The *rnp-6* G281D substitution and overexpression of splicing factors (RNP-6 and SFA-1) lead to sensitivity to infection but promote longevity. Notably, the *rnp-6* G281D mutant is resistant to abiotic stresses, suggesting the sensitivity towards infection is mediated by specialized mechanisms rather than a general decline of fitness in this mutant.

One question that remains to be investigated is the nature of the *rnp-6* G281D mutation. In the context of survival, phenotypically this mutation appears to be a gain of function as both the G281D substitution and RNP-6 over-expression lead to compromised infection survival but prolonged lifespan on OP50. However, the G281D substitution also shows characteristics of a loss of function mutation in other aspects. For example, the transcriptional and splicing changes induced by the G281D substitution are complex with some genes showing patterns similar to that of *rnp-6* RNAi while others having different patterns. We believe that the G281D substitution does not simply increase or decrease the activity of RNP-6 but rather causes a complex change in the affinity of the RNA binding domain to different RNA molecules. The location of the substitution in the RNA binding motif also supports this idea.

One limitation of the splicing assaying with PCR on RNA extracted from whole *C. elegans* animals is that the changes observed might be due to changes in isoform expression pattern in different tissues rather than alternate splicing within the same tissue, as genes can have tissue enriched isoforms. Nevertheless, the *rnp-6* dependence of these splicing changes argues in favor of a true splicing effect.

In mammals, PUF60 is also known as far-upstream element binding protein, which inhibits the transcriptional factor FUSE binding protein (*Duncan et al., 1994*). Although RNP-6 clearly affects the transcriptome, whether a specific transcriptional regulatory function of RNP-6/PUF60 plays a role in immunity is unclear. Often transcriptional changes can reflect the coupling of transcriptional and splicing events (*Bentley, 2014*). In addition, the results from SFA-1 suggest that the effects of RNP-6 on immunity are dependent on the splicing machinery, arguing against a sole involvement of the transcriptional regulatory function of RNP-6/PUF60.

Our data demonstrate that PMK-1 MAPK signaling falls under the control of the splicing machinery. How MAPK signaling is activated in *C. elegans* is not fully understood (*Kim and Ewbank, 2018*).

The current study suggests splicing is an important modulator of MAPK signaling and likely works upstream of TIR-1/SARM. In the non-infected state, splicing factors suppress immunity, but during bacterial infections, splicing is perturbed, leading to the activation of immune responses. Interestingly, our bioinformatic analysis suggests splicing changes upon infection are not global but limited, suggesting specificity. How altering splicing can lead to activation of PMK-1 and immune responses is not well-understood at this point. Splicing is a core cellular process and perturbations of other cellular processes, such as mitochondrial or nucleolar functions, have been shown to activate immunity (*Melo and Ruvkun, 2012*; *Tiku et al., 2018*). Thus, *rnp-6*/PUF60 RNAi derepression of the immune response could reflect damage mediated immunity. On the other hand, *rnp-6* G281D substitution and overexpression exhibit an immunosuppressive function, arguing for a more novel and specific role of splicing in the regulation of immune defense.

In support of the latter idea, MyD88, a Toll-like-receptor signaling adaptor and an analog (functionally similar but differ significantly at the sequence level) of *tir-1* in mammals, is controlled by alternative mRNA splicing (*De Arras and Alper, 2013*). Unlike the situation in mammalian cells, however, we did not observe any significant changes of *tir-1* splicing upon infection or *rnp-6* G281D substitution, nor of other MAPK signaling components. Conceivably other means to regulate *tir-1* are at play. Splicing acts downstream of dietary restriction to promote longevity (*Heintz et al., 2017*). Aberrant activation of PMK-1 causes toxicity and shorter lifespan in *C. elegans* (*Cheesman et al., 2016*). Also, in both dietary restricted animals and *daf-2* mutants, PMK-1 MAPK signaling is inhibited (*Wu et al., 2019b*). One attractive hypothesis is that longevity promoting interventions extend lifespan by suppressing MAPK signaling through splicing.

Our study also reveals a novel and evolutionarily conserved function of RNP-6/PUF60 in regulating immunity in mammalian cells. Although *rnp-6* levels remain stable after infection in *C. elegans*, infection reduces the abundance of PUF60 in various mammalian cells. We hypothesize that infection induced down-regulation of PUF60 might be a recent evolutionary adaptation of the immune system. Alternatively, the activities of RNP-6 may be regulated in *C. elegans* by other ways without affecting its abundance. Indeed, infection induced splicing remodeling is dependent on RNP-6, and knocking-down *rnp-6* affects splicing of genes whose splicing is regulated by infection, suggesting that RNP-6 activity is influenced by infection. Reduction of PUF60 by RNAi is sufficient to induce pro-inflammatory cytokine expression, suggesting that PUF60 reduction induced by bacterial infection is a trigger of inflammation. This intriguing observation raises the question of how perturbation of splicing can activate immune responses. One potential explanation is aberrant splicing may produce immuno-stimulatory RNAs which can be detected by nucleic acid sensing PRRs, such as RIG-I (*Yoneyama et al., 2004*), cGAS/STING (*Mankan et al., 2014*; *Sun et al., 2013*) and PKR (*Feng et al., 1992*; *Onomoto et al., 2012*). Overexpression of PUF60, especially the mutated form, is able to reduce inflammation induced by *S. aureus* infection. The G281D mutation, which we identified in *C. elegans,* analogously augments the immunosuppressing activity of PUF60. These results indicate that the RNA binding motif where the mutation resides plays a crucial role in the control of immunity, probably through interacting with some unknown RNA species. Though both RNP-6 and PUF60 apparently suppress the immune response, the outcomes for survival upon pathogen challenge are different. This may reflect how the balance of proinflammatory and anti-inflammatory molecules ultimately affect survival in different contexts. The E3 ubiquitin ligase ipaH9.8 from *Shigella* was reported to bind another splicing factor U2AF35 and modulate host cytokine production (*Okuda et al., 2005*). Whether PUF60 is also targeted by bacterial factors remains to be investigated.

Hypomorphic mutations of PUF60 are known to cause a rare genetic disease called Verheij syndrome (*Abdin et al., 2018*; *El Chehadeh et al., 2017*; *Verheij et al., 2009*; *Xu et al., 2018*). Our results suggest a reduction of PUF60 activity may trigger inflammation, which could be one of the drivers of the pathology. Further examinations on the patient samples are needed to test this interesting possibility. PUF60 is also implicated in HBV infection (*Sun et al., 2017*) and breast cancer (*Sun et al., 2019*). Whether PUF60 mediated immune regulation plays a role in these diseases awaits investigation. Targeting host splicing can also be an alternative therapeutic approach to antibiotic and multidrug resistant bacterial infection.

Our study paves the way to further explore the intriguing connection between splicing, lifespan and innate immunity in an evolutionarily conserved context. We propose that splicing factors might mediate a tradeoff between immunity and longevity, and further postulate that longevity could have

arisen in some species by limiting inflammatory processes. Our results also raise multiple fundamental questions. How do the splicing factors RNP-6 and SFA-1 work to promote longevity, but compromise survival upon bacterial infection? Does RNP-6/PUF60 regulate immunity by interacting with RNA, and if yes, which RNA(s) is/are involved? How does the G281D mutation change the molecular functions of RNP-6/PUF60? Further studies are needed to precisely elucidate how splicing affects host responses towards infection.

## Materials and methods

### Experimental organism- *C. elegans*

The following strains were used in this study: N2 (wildtype), *rnp-6(dh1124)*, *rnp-6(dh1125)*, *rnp-6(dh1127)*, *hlh-30(tm1978)*, *rnp-6(dh1188)*, *rnp-6(dh1127); wbmEx340[eft-3p::3xFLAG::sfa-1 cDNA:: unc-54 3'UTR]; [myo-3p::mCherry::unc-54 3'UTR]*, *dhEx1132[rnp-6p::rnp-6; myo-2p::GFP]* (RNP-6 OE#1), *dhEx1133[rnp-6p::rnp-6; myo-2p::GFP]* (RNP-6 OE#2), *dhEx1134[rnp-6p::rnp-6; myo-2p:: GFP]* (RNP-6 OE#3), *pmk-1(km25)*, *tir-1(tm3036)*, *pmk-1(km25); dhEx1132[rnp-6p::rnp-6; myo-2p:: GFP]*, *rnp-6(dh1127); pmk-1(km25)*, *daf-16(mu86)*, *wbmEx340[eft-3p::3xFLAG::sfa-1 cDNA::unc-54 3'UTR]; [myo-3p::mCherry::unc-54 3'UTR]*, *rnp-6(dh1188); wbmEx340[eft-3p::3xFLAG::sfa-1 cDNA:: unc-54 3'UTR]; [myo-3p::mCherry::unc-54 3'UTR]* and *pmk-1(km25); wbmEx340[eft-3p::3xFLAG::sfa-1 cDNA::unc-54 3'UTR]; [myo-3p::mCherry::unc-54 3'UTR]*. The animals were maintained at 20°C following standard procedures (Brenner, 1974). For all experiments, synchronization of the animals was done using the egg laying method.

### Experimental organism- Mammalian cell culture

HeLa and MEF cells were obtained from ATCC. RAW264.7 cells were obtained from ECACC. The cells were cultured at 37°C with 5% carbon dioxide in a humid atmosphere. Dulbecco's modified Eagle's medium (DMEM) supplemented with 10% fetal calf serum (FCS) was used as the culturing medium. The cells were tested and are negative for mycoplasma contamination.

### Experimental organism- Bacteria

*Staphylococcus aureus* (strain MW2), *Enterococcus faecalis* (strain ATCC 29212), *Pseudomonas aeruginosa* (strain PA14) and *Escherichia coli* (strain OP50, HT115 and OP50 xu363) were used in this study. Bacteria were maintained in glycerol stocks at −80°C. When needed, bacteria were streaked out on appropriate agar plates. Tryptic soy agar (TSA) plates for *S. aureus*, brain heart infusion (BHI) plates for *E. faecalis* and lysogeny broth (LB) plates for *P. aeruginosa* and *E. coli*. Colonies were picked into designated growth medium. *S. aureus* was grown in tryptic soy broth (TSB) medium. *E. faecalis* was grown in BHI medium. *P. aeruginosa* and *E. coli* were grown in LB medium. 100 µg/mL of ampicillin was supplemented to the plates and cultures of *Escherichia coli* (strain HT115 and OP50 xu363) to maintain the plasmids for RNAi feeding experiments.

### Genome sequencing and genetic mapping

The cold resistant mutants were crossed with Hawaiian CB4856 males. Young adults of the F2 generation were exposed to cold stress, and the survivors were singled and propagated on standard nematode growth medium (NGM) plates seeded with *E. coli* OP50. The cold resistant segregants were then pooled together, and genomic DNA were purified using Gentra Puregene Kit (Qiagen). The pooled DNA was sequenced on an Illumina HiSeq platform (paired-end 150 nucleotide). MiModD software (http://www.celegans.de/en/mimodd) was used to locate the mutations.

### *C. elegans* abiotic stress resistance assays

For all stress assays, synchronized young adult hermaphrodites were used. For cold resistance, worms were incubated at 2°C on standard NGM plates with OP50 for the indicated time. The animals were then allowed to recovered at 20°C for 12–24 hr. After that, their survival was score. For heat resistance, animals on standard NGM plates seeded with OP50 were transferred to 35°C. Survival was monitored every hour. For oxidation stress, worms were placed on NGM plates with OP50 supplemented with 20 mM paraquat. Survival was monitored every 12 hr. Animals that did not respond to gentle touch by a worm pick were interpreted as dead.

### *C. elegans* infection survival experiments

*S. aureus* (Strain MW2), *E. faecalis* (Strain ATCC 29212) and *P. aeruginosa* (Strain PA14) were grown in the appropriate medium at 37°C with gentle shaking overnight. 20 µl of the overnight bacterial cultures was seeded at the center of the relevant agar plates, TSA plates with 10 µg/mL nalidixic acid (NAL) for *S. aureus*, BHI plates with 10 µg/mL NAL for *E. faecalis* and modified NGM (Peptone content 3.5 g/L instead of 2.5 g/L), for *P. aeruginosa*. For full lawn experiments, 100 µl of the bacterial culture was seeded and spread all over the surface of the agar plate. The seeded killing assay plates were allowed to grow overnight at 37°C. On the next day, the plates were left at room temperature for at least 6 hr before the infection experiments. Around 25 synchronized young adult worms were transferred to the plates. three technical replicate plates were set up for each condition. The plates were then incubated at 25°C unless specifically stated. Scoring was performed every 24 hr for *E. faecalis* or every 12 hr for *S. aureus* and *P. aeruginosa*. Worms were scored as dead if the animals did not respond to gentle touch by a worm pick. Worms that crawled off the plates or had ruptured vulva phenotypes were censored from the analysis. All *C. elegans* killing assays were performed three times independently unless otherwise stated. Genotypes were concealed for all *C. elegans* infection survival experiments in order to eliminate any investigator-induced bias. Results of each biological replicate of infection survival and lifespan experiments can be found in *Supplementary file 6*.

### Chemical sterilization of *C. elegans* animals

Animals were sterilized with treatment of 5-Fluoro-2'-deoxyuridine (FUDR). L4 animals were incubated on NGM plates with OP50 and 50 µg/mL of FUDR supplementation at 20°C for 18–24 hr.

### Colony Forming Unit (CFU) Assay in *C. elegans*

Chemically sterilized young adults were infected with full lawn *S. aureus* plates for 24 hr. 10 animals were harvested for each genotype. The animals were washed three times with ice cold M9 buffer with 0.1% trion X-100. After washing, the worms were resuspended in 300 µl M9 buffer with 0.1% trion X-100. An aliquot of the buffer was plated on NAL containing TSA plates to quantify external bacterial counts. Animals were then homogenized by a motor-driven pestle grinder. The homogenates were diluted and plated on NAL containing TSA plates. CFU from the pre-homogenized samples were subtracted from the CFU value after homogenization to calculate the CFU per animal values.

### Lifespan assay

Worms were allowed to grow to young adult stage at 20°C on standard NGM plates with OP50. For each genotype, 120 young adults were then transferred to NGM plates with OP50 supplemented with 10 µM of FUDR. The cultures were continued at either 20°C or 25°C. Survival was monitored at least three times per week. Worms which did not respond to gentle touch by a worm pick were scored as dead and were removed from the plates. Animals that crawled off the plate or have ruptured vulva phenotypes were censored. All lifespan experiments were done three times independently unless otherwise stated. Genotypes were concealed during the lifespan experiments in order to eliminate any investigator-induced bias.

### *C. elegans* sample preparation for RNA analysis

*C. elegans* samples used for PCRs and RNA sequencing were prepared in the following way. Animals were allowed to grow at 20°C on NGM plates seeded with OP50 to age synchronized young adults. After washing with M9 buffer, they were transferred to 10 cm TSA plates supplemented with 10 µg/mL NAL carrying either *S. aureus* or heat-inactivated OP50 bacteria. TSA plates with heat-inactivated OP50 serve as the non-infection control here. The *E. coli* culture was concentrated 10 times before seeding on the plates. 500 µl of *S. aureus* or heat-inactivated E. coli OP50 were spread on the TSA plates, which were then incubated at 37°C for 6 hr. *C. elegans* worms were then transferred to the plates after the plates had equilibrated to room temperature. The animals were incubated on the plates at 25°C. After 4 hr of infection, the worms were washed with M9 buffer for two times before lysis with QIAzol Lysis Reagent (Qiagen).

## RNA extraction and cDNA synthesis

*C. elegans* and mammalian cells were lyzed with QIAzol Lysis Reagent. RNA was extracted using chloroform extraction method. The samples were then purified using RNeasy Mini Kit (Qiagen). Purity and concentration of the RNA samples were assessed using a NanoDrop 2000c (peqLab) equipment. cDNA synthesis was performed using iScript cDNA synthesis kit (Bio-Rad). Standard protocols provided by the manufacturers were followed for all mentioned commercial kits.

## Alternative splicing PCR assay

DreamTaq DNA Polymerases (ThermoFisher) was used to amplify the *prg-2, gyg-1 and tos-1* segments. The primer design for *tos-1* is based on a recent publication (*Ma et al., 2011*). PCR reactions were cycled for 35 cycles with annealing temperature of 57°C. The products were visualized by staining with Roti-GelStain (Carl Roth) after agarose gel electrophoresis. For primer sequences, please refer to *Supplementary file 1*.

## Quantitative PCR (qPCR)

Power SYBR Green master mix (Applied Biosystems) was used for qPCR experiments. A JANUS automated workstation (PerkinElmer) was used for pipetting the reagents and cDNA samples into a 384 well plate. Thermal cycling was performed using a ViiA7 384 Real- Time PCR System machine (Applied Biosystems). *snb-1* was used for internal normalization for *C. elegans* targets. GAPDH or RPL13A were used for internal normalization for human targets. GAPDH was used for internal normalization of mouse targets. Relative expression levels were calculated using the comparative CT method. Special attention was paid to ensure that the primer sets detect all annotated isoforms. For primer sequences and binding locations, please refer to *Supplementary file 1*, *2*, *3* and *4*.

## Transcriptomic profiling and bioinformatic analysis

1 μg of total RNA was used per sample for library preparation. We used the protocol of Illumina Tru-Seq stranded RiboZero. After purification and validation (2200 TapeStation; Agilent Technologies), total of 12 libraries (non-infected N2, non-infected *rnp-6(dh1127)*, infected N2 and infected *rnp-6 (dh1127)* in each biological replicate, three biological replicates in total) were pooled for quantification using the KAPA Library Quantification kit (Peqlab) and the 7900HT Sequence Detection System (Applied Biosystems). The libraries were then sequenced on one lane of an Illumina HiSeq4000 sequencing system using a paired end 2 × 75 nt sequencing protocol. For data analysis, Wormbase genome (WBcel235_89) was used for alignment of the reads. This was performed with the Hisat version 2.0.4 software. Differentially expressed genes (DEGs) (q-value <0.05) between different samples were identified using the stringtie version 1.3.0, followed by Cufflinks version 2.2. The DAVID (Database for Annotation, Visualization and Integrated Discovery) database was used for enrichment and Gene Ontology (GO) analysis. Cufflinks, KISSDE and SAJR were used for the splicing analysis. q value/p adjusted <0.05 is considered to be significant.

## Protein sample preparation and western blotting

For *C. elegans* samples, two different methods were employed. Animals were picked into ice-cooled M9 buffer. 6X Laemmli lysis buffer with 5% 2-mercaptoethanol was then added to the mixture, which was then frozen in liquid nitrogen immediately. Alternatively, animals were first washed with M9 buffer. The worm pellets were resuspended in RIPA buffer supplemented with cOmplete Protease Inhibitor (Roche) and PhosSTOP (Roche) and snap frozen in liquid nitrogen. The thawed samples were lyzed using Bioruptor Sonication System (Diagenode). For mammalian cell samples, the cells were first washed with PBS and then lyzed with RIPA buffer supplemented with cOmplete Protease Inhibitor (Roche) and PhosSTOP (Roche). Cell debris was removed by centrifugation. Protein concentration was estimated using Pierce BCA Protein Assay Kit (Thermo Fisher Scientific). Protein samples were then heated to 95°C for 10 min in Laemmli buffer with 0.8% 2-mercaptoethanol in order to denature the proteins. The samples were sonicated in a sonicating water bath for 10 min if there is excessive DNA, which obscures the subsequent loading process. Samples were loaded on 4–15% MiniPROTEAN TGXTM Precast Protein Gels (Bio-Rad), and electrophoresis was performed at constant voltage of 200V for around 40 min. After separation, the proteins were transferred to PVDF membranes using Trans-Blot TurboTM Transfer System (BioRad). 5% Bovine serum albumin (BSA) or

5% milk in Tris-buffered Saline and Tween20 (TBST) were used for blocking of the membranes. After antibody incubations and washing with TBST buffer, imaging of the membranes was performed with ChemiDoc Imager (BioRad). Western Lightning Plus Enhanced Chemiluminescence Substrate (PerkinElmer) was used as the chemiluminescence reagent. A list of antibodies is provided in *Supplementary file 5*.

## Co-immunoprecipitation

Worms expressing HA::RNP-6, FLAG-SFA-1 or both were harvested, and proteins were extracted using the Bioruptor method. We used buffer containing 0.5% NP40, 150 mM NaCL and 50 mM Tris pH 7.4 supplemented with cOmplete Protease Inhibitor (Roche) and PhosSTOP (Roche) for immunoprecipitation. Flag immunoprecipitation was performed using Dynabeads Protein G (ThermoFisher Scientific) and FLAG M2 mouse monoclonal antibody (Sigma), following manufacturer's protocols. Proteins were eluted from the beads by boiling with Laemmli buffer.

## RNAi in *C. elegans*

Two different strains of RNAi competent bacterial strains were used in this study, namely *E. coli* HT115 and *E. coli* OP50 xu363. The HT115 bacteria were from the Vidal or Ahringer library. OP50 xu363 bacteria were transformed with dsRNA expression plasmids, which were extracted from the respective HT115 bacterial strains. The bacteria were grown in LB medium supplemented 100 μg/mL ampicillin at 37°C overnight with gentle shaking. The culture was spread on RNAi plates, which are NGM plates with addition of 100 μg/mL ampicillin and 0.4 mM isopropyl β-D-1-thiogalactopyranoside (IPTG). The RNAi bacteria were allowed to grow on the plates at room temperature for at least 12 hr. RNAi was initiated by letting the animals to feed on the desired RNAi bacteria. For infection survival experiments, *rnp-6* RNAi was initiated around 30 hr before infection in order to prevent developmental abnormalities.

## Transfection of mammalian cells

siRNA against PUF60 and non-targeting siRNA were purchased from Dharmacon (GE Healthcare Life Sciences). Lipofectamine 3000 (ThermoFisher Scientific) was used as the transfection reagent for both siRNA and plasmids in HeLa cells. Lipofectamine RNAiMAX was used for siRNA transfections in RAW264.7 cells. For all transfection experiments, we followed the manufacturers' protocols.

## Infection of mammalian cells

*S. aureus* (MW2) was used for infection at multiplicity of infection (MOI) 100. Late logarithmic phase grown bacterial *S. aureus* cultures were diluted in DMEM with 10% FCS to the designated MOI. Cells were washed with PBS and then incubated with the bacterial suspension for 10 min at room temperature and another 30 min at 37°C with 5% $CO_2$ in a humid atmosphere to allow internalization of the bacteria. After that, the infection medium was removed, and the residual extracellular bacteria were removed by washing with PBS and incubation with DMEM medium with 10% FCS and 50 μg/ml gentamicin for 1 hr. Lower concentration of gentamicin (10 μg/ml) was used for continued culture after the initial 1 hr incubation.

To quantify the internalized bacteria, the cells were lyzed with PBS containing 0.3% of Triton X-100 at room temperature for 5 min to release intracellular bacteria. Amounts of bacteria were determined by serial dilution and plating on TSA plates.

Cell viability was assayed by trypan blue exclusion method. At the indicated time points, infected HeLa cells were first washed with PBS to remove any dead cells in suspension. The cells were then dissociated by incubation with 0.5% trypsin. Cells were then mixed with trypan blue solution. Live cells which do not uptake trypan blue were counted using a hemocytometer.

## Materials availability

Requests for reagents, *C. elegans* strains, or any other resources should be directed to and will be fulfilled by the Lead Contact, Adam Antebi (Aantebi@age.mpg.de).

## Acknowledgements

We thank the Caenorhabditis Genetics Center for providing the *C. elegans* strains. We would also like to acknowledge Cologne Center for Genomics for conducting the RNA sequencing experiments. We especially thank the Bioinformatics Core Facility at Max Planck Institute for Biology of Ageing (Dr. Jorge Boucas, Dr. Rafael Cuadrat and Dr. Franziska Metge) for assisting with the bioinformatic analysis. The wildtype PUF60 expression plasmid was provided by Dr. Kenji Nakashima at Hamamatsu University School of Medicine. The rabbit anti-total-PMK-1 antibody was provided by Read Pukkila-Worley lab, UMass Medical School. This work was supported financially by Max Planck Society.

## Additional information

### Funding

| Funder | Author |
| --- | --- |
| Max-Planck-Gesellschaft | Chun Kew<br>Wenming Huang<br>Adam Antebi |

The funders had no role in study design, data collection and interpretation, or the decision to submit the work for publication.

### Author contributions

Chun Kew, Conceptualization, Resources, Data curation, Formal analysis, Validation, Investigation, Methodology, Writing - original draft; Wenming Huang, Conceptualization, Resources, Data curation, Formal analysis, Investigation, Methodology, Writing - review and editing; Julia Fischer, Raja Ganesan, Investigation, Methodology, Writing - review and editing; Nirmal Robinson, Methodology, Writing - review and editing; Adam Antebi, Conceptualization, Resources, Formal analysis, Supervision, Funding acquisition, Project administration, Writing - review and editing

### Author ORCIDs

Chun Kew  https://orcid.org/0000-0003-0311-5092
Wenming Huang  https://orcid.org/0000-0001-5252-3928
Nirmal Robinson  https://orcid.org/0000-0002-7361-9491
Adam Antebi  https://orcid.org/0000-0002-7241-3029

### Decision letter and Author response

Decision letter https://doi.org/10.7554/eLife.57591.sa1
Author response https://doi.org/10.7554/eLife.57591.sa2

## Additional files

### Supplementary files

- Supplementary file 1. List of primers.
- Supplementary file 2. qPCR primer binding sites for *C. elegans* targets.
- Supplementary file 3. qPCR primer binding sites for human targets.
- Supplementary file 4. qPCR primer binding sites for mouse targets.
- Supplementary file 5. List of antibodies.
- Supplementary file 6. Table of survival data.
- Transparent reporting form

### Data availability

RNA-seq data has been deposited in GEO. Accession code GSE141097.

The following dataset was generated:

| Author(s) | Year | Dataset title | Dataset URL | Database and Identifier |
|-----------|------|---------------|-------------|-------------------------|
| Kew C, Antebi A | 2019 | Evolutionarily Conserved Regulation of Immunity by the Splicing Factor RNP-6/PUF60 | https://www.ncbi.nlm.nih.gov/geo/query/acc.cgi?acc=GSE141097 | NCBI Gene Expression Omnibus, GSE141097 |

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
