## [Decision Letter]

**Acceptance summary:**

The authors identified a splicing factor RNP-6/PUF60 whose activity suppresses immunity, but also promotes longevity, suggesting a trade-off between these two processes. The authors reveal the role of specific components of the splicing machinery in the modulation of immunity, which is important because the role of splicing in immunity is still poorly characterized. The merit of this article is that it illuminates a novel aspect of innate immune regulation in the nematode *Caenorhabditis elegans* that might be conserved in other animals.

**Decision letter after peer review:**

[Editors’ note: the authors submitted for reconsideration following the decision after peer review. What follows is the decision letter after the first round of review.]

Thank you for submitting your work entitled "Evolutionarily Conserved Regulation of Immunity by the Splicing Factor RNP-6/PUF60" for consideration by *eLife*. Your article has been reviewed by three peer reviewers, and the evaluation has been overseen by a Reviewing Editor and a Senior Editor. The following individual involved in review of your submission has agreed to reveal their identity: Orane Visvikis (Reviewer #1).

Our decision has been reached after consultation between the reviewers. As you can see from the comments, the reviewers were rather positive about the manuscript and the possibility to publish it in *eLife*. Nevertheless, they feel that there are too many requested work to allow a revision in two months. The consensus was therefore to reject the paper but to allow resubmission. If you decide to submit a revised version, it will be considered as a new submission but will likely be treated by the same reviewers. The decision 'Reject but allowing resubmission' gives you more time to revise the manuscript. We therefore encourage you to submit a new revised version of your manuscript that carefully addresses the point raised by the reviewers. If you feel that you cannot address the reviewer's comments, it is also possible to transfer the manuscript to another journal.

Reviewer #1:

In this article, Kew et al. have used *C. elegans* model and mammalian cell lines to elegantly unravel the evolutionary conserved immune function of the splicing factor RNP6/PUF60. By combining numerous and complementary approaches (genetic screen, CRISPR gene editing, RNA-sequencing, in vivo survival assays, molecular and biochemical analysis), the authors provide strong evidence of the repressive immune function of RNP-6/PUF60 in *C. elegans*, relying both on control of RNA splicing and gene expression, the latter lying upstream the *pmk-1*/p38 pathway. They also provide evidence that another splicing factor, namely SFA-1, work with RNP-6 to mediate host defense, underlying the importance of the RNA splicing function innate immunity. Finally, they show that PUF60 expression is repressed during infection of mammalian cell lines, which increases pro-inflammatory cytokines expression and promotes survival of infected cells. Overall, this well-conducted study reveals a novel function of PUF60 and contribute to enlarge the knowledge of a field in expansion. I would recommend this article for publication in *eLife*.

1) A major concern of this study relays on how survival assay were performed (partial or full lawn) which is not clear and may be of critical importance for the conclusion. Figure 4A: this survival assay testing *rnp-6* RNAi was apparently performed using plates with partial lawn of *S. aureus* as the authors state to confirm this result with a survival assay using full lawn plates (subsection “The activity of RNP-6 impairs immunity”). Is the survival assay with partial lawn of *S. aureus* their standard assay ? This need to be clarified in the next and in the Materials and methods. Indeed, it is not clear how the experiments with *rnp-6(dh1127)* and *rnp-6(dh1125)* in Figure 1H (as well as in Figure 1I-J with other pathogens) and Figure 1—figure supplement 1I, J and K, or with the overexpressing OE strains (Figure 4D) were performed. Indeed, if based on the full lawn assay, the authors can conclude that the *rnp-6* RNAi phenotype, which display extend survival compared to control, is not a difference in bacterial exposure, it is not clear whether the phenotypes of rpn-6 G281D mutants or of the OE strains, which display enhanced susceptibility to pathogens (Esp) compared to control, are due to reduced immune response or reduced aversive behavior (and increased bacterial exposure) compared to N2. Full lawn experiment should be performed in order to conclude on the immune role of *rnp-6*. If partial lawn have been performed, the authors need to compare the bacterial lawn occupancy of RNP-6D281R mutants and the OE strains with N2. The author will be able to conclude on the immune regulatory function of RNP-6 only if RNP-6D281R mutant and N2 have the same avoidance behavior (and been exposed in a similar way to *S. aureus*).

2) As a splicing factor, rpn-6 affects RNA splicing. I believe it would improve the manuscript to specify how primer pairs have been designed so that qPCR experiments reveal an actual change in gene expression and not a change in RNA splicing (Figure 2A).

3) Figure 5GH and Figure 6H: condition of these experiments are rather surprising. While most of the *pmk-1* and *rnp-6* genetic experiment are made in animals infected by *S. aureus*, the authors analyze the effect RNP6 on PMK-1 phosphorylation either in resting condition (RNAi) or during infection by *P. aeruginosa*. It would greatly increase the quality of the conclusions to show 1) the level of P-PMK1 in N2i and *rnp-6i* animals infected by *S. aureus*; 2) the level of P-PMK1 in N2, *rnp-6(dh1127)* and RNP6 OE strain in the context of *S. aureus* infection. In addition, if anti-pmk1 (Kim et al., 2002) can be obtained, it would be good to blot total *pmk-1* to robustly show that variation of P-*pmk-1* are really changes in signaling rather a change in total *pmk-1* protein level.

4) I believe the author should cite the work of the Sasakawa group (Okuda et al., 2005) who had previously identified the Shigella NEL effector IpaH9.8 to bind RNA splicing factor U2AF35 to modulate host immune response. Although IpaH9.8 had later been shown to also target NEMO to control NF-κB and host immune response (Ashida et al. Nat Cell Biol 2010), it was shown that U2AF35 siRNA treatment reduces pro-inflammatory cytokines in Shigella-infected cells HeLa (Okuda et al., 2005). Although this paper show opposite results in term of cytokine expression, it is a first evidence that modulation of a splicing factor, which is related to PUF60, during bacterial infection impacts immune responses, which reinforce the findings of Kew et al.

Reviewer #2:

This is a very solid investigation of how splicing factors influence stress and longevity signaling in *C. elegans* that include experiments showing similar interactions in human cells. The role of RNA splicing in stress responses of animal cells is poorly studied. This study identifies a conserved RNA binding protein that plays important roles in stress responses of *C. elegans* and human cells. Precise molecular mechanism have not been revealed, but the role of *rnp-6* and splicing in immune response are convincingly demonstrated. Furthermore, the experiments are well conceived and reported. The genetic controls are particularly robust.

1) Stress survival and longevity data – These assays can be notoriously variable between trials. Therefore, reporting of data from at least three independent trials is customary for these types of studies. The comparisons should be analyzes as individual trials as well as combined. These data can be provided in supporting files.

2) I have concerns over the use of the *ret-1* reporter system. It appears that the GFP and RFP are largely expressed is different tissues (head and intestine, respectively) and that the changes in ratio are largely attributed to changes in only GFP versus a true shift from one to the other. The authors should include data for changes in GFP and RFP in addition to the rations. Furthermore, controls for total expression of the reporter would be needed to be sure that these effects are not a reflection of changes in total expression in certain tissues, versus a true shift in splicing within common tissues.

3) Figures 5G-H and 6H – quantification of the representative data for Figure 5H is missing. Also, it is customary to include blots for total PMK-1 to determine if the effects are at the total level or protein modification level.

Reviewer #3:

Kew et al. investigate the link between splicing and the immunity and longevity in *C. elegans* and extend their study to a mammalian cell culture model. Because of my expertise, I restrict my comments on the experiments to those using *C. elegans*.

1) While the results are intriguing and would be of broad interest, I was concerned by Figure 3F, as in the infected sample, there are clearly larvae hatched inside that have a high GFP level and low mCherry level. This is a real problem. It indicates that (i) the experiments are not properly conducted (ii) the results involving differential splicing could be a result of differences that exist between different stages. Even if the worms shown were exceptional, it is notable that a significant number of the 83 genes alternatively spliced upon *S. aureus* infection in wildtype animals are associated with aldicarb resistance. As alternative splicing is tissue as well as stage specific (e.g. neuron-specific isoforms for many genes), this raises the possibility that the results are due to a differential effect of infection on transcription in a single tissue (including through a general effect on cell viability or developmental speed). The same applies for the effect seen in the *rnp-6* mutant; changes in transcription within one tissue, or even subtle changes in developmental stage, would give the same result as alterations in splicing across tissues in an age-synchronized population.

2) Given the apparent strong effect on the ret reporter, which presumably should reflect endogenous splicing patterns, how is it that *ret-1* was not identified in the sequencing-based analysis? It is unfortunate that the authors did not generated this type of reporter for at least one of the genes they identified as undergoing differential splicing.

3) Figure 5—figure supplement 1 shows sequencing reads coverage for several genes. If the figure is only matching pairs for their paired end 2x75nt sequencing, which it should be for this type of analysis, the level of intronic signal for several genes is unexpected. So, when they write, "The splicing patterns of these genes are not significantly affected by *S. aureus* infection or the *rnp-6* G281D mutation", what is the criterion for "significant", especially if one looks at the last example, *cebp-1*, for which there is a stark difference regarding retention of the last intron between wild-type and *rnp-6* samples. [typo: *S. aurues* in figure]. This is particularly relevant given the known links between *cebp-1*, its 3' UTR (pubmed/28673818), *pmk-1* (pubmed/27927209) and innate immunity.

4) To understand the basis of altered resistance, it is standard in the field to perform simple tests to assay the extent of bacterial colonization.

5) Interpretation of Figure 2E is complicated by the fact that the constitutive expression level of at least one of the genes under study differs between wild-type and mutant worms. Thus, for *nlp-30*, there is the same fold-change after infection in mutant and wild-type, so the induction is not *rnp-6* dependent. One therefore wonders whether their strategy for gene selection (Figure 2C) is entirely appropriate.

[Editors’ note: further revisions were suggested prior to acceptance, as described below.]

Thank you for submitting your article "Evolutionarily conserved regulation of immunity by the splicing factor RNP-6/PUF60" for consideration by *eLife*. Your article has been reviewed by three peer reviewers, and the evaluation has been overseen by a Reviewing Editor and Tadatsugu Taniguchi as the Senior Editor. The following individual involved in review of your submission has agreed to reveal their identity: Orane Visvikis (Reviewer #1).

The reviewers have discussed the reviews with one another and the Reviewing Editor has drafted this decision to help you prepare a revised submission.

Summary:

The authors identified a splicing factor RNP-6/PUF60 whose activity suppresses immunity, but promotes longevity, suggesting a trade-off between these two processes. The authors reveal the role of specific components of the splicing machinery in the modulation of immunity, which is important because the role of splicing in immunity is still poorly characterized. The merit of this article is that it illuminates a novel aspect of innate immune regulation in the nematode *Caenorhabditis elegans* that might be conserved in other animals.

Essential revisions:

1) Despite the additional data provided, it is still not clear how RNP-6 impacts the PMK-1 pathway during *S. aureus* and *P. aeruginosa* infection. "RNP-6 exerts its inhibitory effects on immunity by suppressing the activity of PMK-1" (L 305-306) is over-stated as the authors do not have proof that *rnp-6* impact the activation (phosphorylation) of PMK-1 during infection. Indeed:

– The authors find no effect of *rnp-6* OE strain or *rnp-6* G281D mutant on the phospho-level of PMK-1 during *S. aureus* infection (rebuttal experiment).

– The authors state that *rnp-6* RNAi reduces P-PMK-1 during *S. aureus* infection, but this result is rather questionable considering the western-blot shown (rebuttal). There are huge differences in total protein levels as revealed by the loading control (actin), but these differences are not as strong in total PMK-1. This either suggest that 1) that the level of total PMK-1 varies, or 2) the experiment is not well conducted.

– Figure 5—figure supplement 1C: control of total PMK-1 has not been provided. In addition, the quantification of this blot clearly indicates that there is a huge variation between experiments, and that P-PMK1 variation between conditions is not statistically significant.

I don't think this affect the quality of the manuscript, but the author should be accurate in their interpretation, and the their conclusion modified accordingly (subsection “RNP-6 inhibits PMK-1 MAPK signaling”). Similarly, the text should be changed in the Discussion "How altering splicing can lead to activation of PMK-1.…".

2) As requested, the authors have performed bacterial lawn occupancy experiments and found it is increased in *rnp-6* G281D mutant compared to WT, which still leaves a doubt on a potential lack of avoidance behavior of the *rnp-6* mutant (rebuttal). Fortunately, the author also provide full lawn survival experiments as requested and found the *rnp-6* G281D and *rnp-6* OE strains to also display Esp compared to WT animals, indicating that these strains indeed have an immune defect. The authors keep describing this immune mechanism as "resistance" which I believe is not an appropriate term, especially in this new version of the manuscript where the authors, to answer reviewer 3, now assess the bacterial burden. Resistance, as defined by the capacity to limit bacterial burden, is opposed to tolerance, defined as ability to limit health impact of a given pathogen burden (see Schneider and Ayres Nat Rev Immunol 2008, Medzhitov, Schneider and Soares, Science 2012). In this type of survival assays, where animals are constantly exposed and fed with pathogenic bacteria, it is difficult to assess the resistance capacity. Assessing resistance capacity is easier to detect when animals are transfer onto non-pathogenic food (Sifri et al., Infect Immun 2003). Accordingly, the colonization assays, performed on animals constantly fed for 24h on *S. aureus*, show similar bacterial burden between control and *rnp-6*G281D mutant, *rnp-6* OE strains or *rnp-6* RNAi treated animals (Figure 1—figure supplement 1E, Figure 4—figure supplement 1B-H). Rather, these full lawn survival assays, controlled by similar colonization assays, allow to assess the tolerance capacity of the strains. Together, these results indicate that the immune function of *rnp-6* relies on a tolerance mechanism. This should be indicated in the text which, I believe, would improve the quality of the article. However, the term resistance, which has been repeatedly found though-out the manuscript, is miss-used and should be replaced by i.e.: "decreased sensitivity" or "increased survival" and only when appropriate, by tolerance. Whether *rnp-6* has also resistance immune branch should not be excluded, as it was difficult to assess in this experimental setting. Please also correct : "*rnp-6* RNAi did not affect *S. aureus* bacterial burden, suggesting the resistance is caused by independent mechanisms".

3) There are still problems with the data presented in Figure 3 and in Figure 3—figure supplement 1. Indeed, there is no demonstration of a correlation between splicing as measured by PCR and using reporter genes. Thus, they neither "generate this type of reporter for at least one of the genes they identified as undergoing differential splicing", nor do they provide PCR evidence to support their observations with the *ret-1* reporter. As mCherry and GFP have different half-lives, changes in fluorescence intensity under a pathological situation need not reflect a change in splicing. The magnitude of the change of GFP expression is considerably higher than that of mCherry; it seems that a change in splicing is not the most important effect here. Further, according to the dot plot, the GFP level for the vast majority of "Infected *rnp-6*" worms is below that for the majority of "*rnp-6(dh1127)*" worms, so the photomicrograph is totally unrepresentative and very misleading.

4) Statements like, "By contrast, we observed that the infection sensitive G281D substitution mutant of RNP-6 had a consistently suppressed phosphorylation levels of PMK-1 under *P. aeruginosa* infection condition in multiple biological replicates, although this did not reach statistical significance due variability of phospho-PMK-1 levels between replicates" are not appropriate. The results are simply not statistically significant and no inferences can be made.

---

## [Author Response]

[Editors’ note: the authors resubmitted a revised version of the paper for consideration. What follows is the authors’ response to the first round of review.]

Reviewer #1:[…] 1) A major concern of this study relays on how survival assay were performed (partial or full lawn) which is not clear and may be of critical importance for the conclusion. Figure 4A: this survival assay testing rnp-6 RNAi was apparently performed using plates with partial lawn of *S. aureus* as the authors state to confirm this result with a survival assay using full lawn plates (subsection “The activity of RNP-6 impairs immunity”). Is the survival assay with partial lawn of *S. aureus* their standard assay ? This need to be clarified in the next and in the Materials and methods. Indeed, it is not clear how the experiments with rnp-6 (dh1127) and rnp-6 (dh1125) in Figure 1H (as well as in Figure 1I-J with other pathogens) and Figure 1—figure supplement 1I, J and K, or with the overexpressing OE strains (Figure 4D) were performed. Indeed, if based on the full lawn assay, the authors can conclude that the rnp-6 RNAi phenotype, which display extend survival compared to control, is not a difference in bacterial exposure, it is not clear whether the phenotypes of rpn-6 G281D mutants or of the OE strains, which display enhanced susceptibility to pathogens (Esp) compared to control, are due to reduced immune response or reduced aversive behavior (and increased bacterial exposure) compared to N2. Full lawn experiment should be performed in order to conclude on the immune role of rnp-6. If partial lawn have been performed, the authors need to compare the bacterial lawn occupancy of RNP-6D281R mutants and the OE strains with N2. The author will be able to conclude on the immune regulatory function of RNP-6 only if RNP-6D281R mutant and N2 have the same avoidance behavior (and been exposed in a similar way to *S. aureus*).

Although we used partial lawn for our standard assays, we also performed the infection survival experiments using full lawn *S. aureus* plates.

**Author response image 1. sa2fig1:** 

Clearly, both *rnp-6(dh1127)* mutants and RNP-6 overexpressing animals exhibit Esp on full lawn *S. aureus* infection plates, indicating that the Esp phenotype cannot be explained by differential exposure to bacteria.We also compared the lawn occupancy of *rnp-6(dh1127)* mutants to N2. We observed a general increase of lawn occupancy in all strains during the course of *S. aureus* infection, which is probably due to physical inactivity caused by advanced infection. Interestingly, *rnp6(1127)* mutants had a higher lawn occupancy compared to N2. This could be explained by the more rapid progression of infection in this strain. Another infection sensitive mutant strain, *pmk-1(km25)*, displayed a similar phenotype, suggesting higher lawn occupancy may be a common phenotype of infection sensitive strains as their physical activity declines faster due to quicker progression of the infection process.

All in all, we conclude that differential exposure to bacteria is not the cause of the Esp phenotype of *rnp-6(dh1127)* and RNP-6 overexpressing animals.

2) As a splicing factor, rpn-6 affects RNA splicing. I believe it would improve the manuscript to specify how primer pairs have been designed so that qPCR experiments reveal an actual change in gene expression and not a change in RNA splicing (Figure 2A).

Special attention was paid to ensure that the qPCR primer binding sites are located in regions which are included in all annotated isoforms. Therefore, the qPCR primers should detect all annotated isoforms and reflect an actual change in gene expression. We provide visualization of all qPCR primer binding sites in the Supplementary file 1. Notes have also been added to the Materials and methods section to clarify this issue.

While we were preparing the lists and the schematics of the primers, we found that the primers for *Y43F8B.9* was mislabeled as *frm-7*. We corrected the corresponding figure in Figure 2—figure supplement 1. This does not affect the conclusion of the study.

3) Figures 5G-H and 6H: condition of these experiments are rather surprising. While most of the pmk-1 and rnp-6 genetic experiment are made in animals infected by *S. aureus*, the authors analyze the effect RNP6 on PMK-1 phosphorylation either in resting condition (RNAi) or during infection by P. aeruginosa. It would greatly increase the quality of the conclusions to show 1) the level of P-PMK1 in N2i and rnp-6i animals infected by S. aureus; 2) the level of P-PMK1 in N2, rnp-6(dh1127) and RNP6 OE strain in the context of *S. aureus* infection. In addition, if anti-pmk1 (Kim et al., 2002) can be obtained, it would be good to blot total pmk-1 to robustly show that variation of P-pmk-1 are really changes in signaling rather a change in total pmk-1 protein level.

We added the total PMK-1 blots for the *rnp-6* and *sfa-1* RNAi Western blots (see Figures 5G and 6H).

We also measured the relative levels of phospho-PMK-1 in *S. aureus* infected animals.

Generally, we did not observe any significant effects of *S. aureus* on the levels of phosphoPMK-1. However, we observed a reduction of phospho-PMK-1 upon *S. aureus* infection in *rnp-6* RNAi treated animals. It seems that the PMK-1 activity is affected by *S. aureus* in an RNP-6 dependent manner.

**Author response image 3. sa2fig3:** 

**Author response image 4. sa2fig4:** 

4) I believe the author should cite the work of the Sasakawa group (Okuda et al., 2005) who had previously identified the Shigella NEL effector IpaH9.8 to bind RNA splicing factor U2AF35 to modulate host immune response. Although IpaH9.8 had later been shown to also target NEMO to control NF-κB and host immune response (Ashida et al. Nat Cell Biol 2010), it was shown that U2AF35 siRNA treatment reduces pro-inflammatory cytokines in Shigella-infected cells HeLa (Okuda et al., 2005). Although this paper show opposite results in term of cytokine expression, it is a first evidence that modulation of a splicing factor, which is related to PUF60, during bacterial infection impacts immune responses, which reinforce the findings of Kew et al.

The reference has been added.

Reviewer #2:[…] 1) Stress survival and longevity data – These assays can be notoriously variable between trials. Therefore, reporting of data from at least three independent trials is customary for these types of studies. The comparisons should be analyzes as individual trials as well as combined. These data can be provided in supporting files.

We provide the p values, number of animals analysed and median survival of all replicates of all infection survival and lifespan experiments in Supplementary file 3.

2) I have concerns over the use of the ret-1 reporter system. It appears that the GFP and RFP are largely expressed is different tissues (head and intestine, respectively) and that the changes in ratio are largely attributed to changes in only GFP versus a true shift from one to the other. The authors should include data for changes in GFP and RFP in addition to the rations. Furthermore, controls for total expression of the reporter would be needed to be sure that these effects are not a reflection of changes in total expression in certain tissues, versus a true shift in splicing within common tissues.

First, we quantified the GFP and mCherry signals from the anterior gut as this is the relevant tissue for bacterial infection in *C. elegans*. Therefore, the quantification reflects a shift of abundance of GFP and mCherry in the same tissue, not a comparison overall.

As the reviewer requested, we provide the quantification of GFP and mCherry separately. Please note that we repeated the infection *ret-1* reporter experiment using a new protocol as suggested by reviewer 3 (with addition of FUDR treatment). The data shown below is the new data.

**Author response image 5. sa2fig5:** 

*S. aureus* infection clearly reduced the GFP to mCherry ratio, and this effect was partially suppressed by the *rnp-6(dh1127)* mutation. GFP intensity decreased drastically in infected N2 but only moderately in infected *rnp-6(dh1127)* animals. mCherry expression was induced by *S. aureus* infection in both WT and *rnp-6(dh1127)* animals. Although we observed a trend of lower mCherry signal in infected *rnp-6(dh1127)* mutants (mean=11.944) compared to infected N2 (mean=12.499), the effect does not reach statistical significance.Similar results can also be seen in *rnp-6* RNAi treated animals. knocking-down *rnp-6* reduced GFP to mCherry ratio, which mimics the situation of infection. *rnp-6* RNAi not only reduced the GFP intensity, but also induced the mCherry signal.

**Author response image 6. sa2fig6:** 

**Author response image 7. sa2fig7:** 

Based on these data, we conclude that both bacterial infection and *rnp-*6 reduction result in a similar shift of the *ret-1* splicing reporter, which is caused by a decrease of GFP expression and a concomitant increase of mCherry.

3) Figures 5G-H and 6H – quantification of the representative data for 5H is missing. Also, it is customary to include blots for total PMK-1 to determine if the effects are at the total level or protein modification level.

We did the quantification of phospho-PMK-1 for *P. aeruginosa* infected animals. Although we observed a consistent trend of PMK-1 activation upon *P. aeruginosa* infection in three independent replicates, the relative levels of activation were highly variable and did not reach statistical significance. We moved this data to the figure supplement.

**Author response image 8. sa2fig8:** 

Further, we added the total PMK-1 blots for the *rnp-6* and *sfa-1* RNAi Western blots (see Figures 5G and 6H).

Reviewer #3:Kew et al. investigate the link between splicing and the immunity and longevity in *C. elegans* and extend their study to a mammalian cell culture model. Because of my expertise, I restrict my comments on the experiments to those using *C. elegans*.1) While the results are intriguing and would be of broad interest, I was concerned by Figure 3F, as in the infected sample, there are clearly larvae hatched inside that have a high GFP level and low mCherry level. This is a real problem. It indicates that (i) the experiments are not properly conducted (ii) the results involving differential splicing could be a result of differences that exist between different stages. Even if the worms shown were exceptional, it is notable that a significant number of the 83 genes alternatively spliced upon *S. aureus* infection in wildtype animals are associated with aldicarb resistance. As alternative splicing is tissue as well as stage specific (e.g. neuron-specific isoforms for many genes), this raises the possibility that the results are due to a differential effect of infection on transcription in a single tissue (including through a general effect on cell viability or developmental speed). The same applies for the effect seen in the rnp-6 mutant; changes in transcription within one tissue, or even subtle changes in developmental stage, would give the same result as alterations in splicing across tissues in an age-synchronized population.

The worms were synchronized with special care, as *rnp-6(dh1127)* has a slight developmental delay. We did the egg lay sequentially to ensure the synchronization of different strains. Animals were also examined visibly before experiments to make sure that they are well-synchronized.

To further exclude the possibility of the effects of staging, we repeated the splicing PCR of *prg-2*, *gyg-1* and *tos-1* on day2 adult animals.

**Author response image 9. sa2fig9:** 

The splicing patterns of day2 adults are generally consistent with those of day 1 young adults, except for *gyg-1*, for which day 2 *rnp-6(dh1127)* animals responded to infection comparably to N2 animals. In day 1 young adults, the *gyg-1* splicing change induced by *S. aureus* infection is partially suppressed in *rnp-6(dh1127)* animals. Because of the special attention given for the synchronization of the animals and the reproducibility of the results in another life stage (Day 2 adulthood), the splicing changes we observed are unlikely to be caused by lack of synchronization or subtle changes in developmental stage, but rather to be the results of infection and/or the *rnp-6* mutation.We also repeated the *ret-1* reporter experiment with FUDR sterilized animals, which eliminates hatching of larvae, and we obtained similar results to the experiments performed without FUDR. We quantified the signals from anterior gut as this is the relevant tissue for bacterial infection in *C. elegans*. See Author response image 5.

Given the tight staging, it seems less likely that tissue composition is different between genotypes. If there are tissue specific effects (e.g. differential alternative splicing in the nervous system) it still would be due to splicing, and this is inherently interesting, but lies beyond the scope of the study.

2) Given the apparent strong effect on the ret reporter, which presumably should reflect endogenous splicing patterns, how is it that ret-1 was not identified in the sequencing-based analysis? It is unfortunate that the authors did not generated this type of reporter for at least one of the genes they identified as undergoing differential splicing.

First, the RNA sequencing and the *ret-1* reporter imaging were done at different timepoints. We collected the RNA at post-infection 4 hours whereas the imaging of *ret-1* reporter was done at post-infection 9 hours. The effects on *ret-1* maybe a delayed response. To test this hypothesis, we repeated the *ret-1* reporter imaging experiment at post-infection 4 hours.

**Author response image 10. sa2fig10:** 

At this earlier timepoint, the effects on the *ret=*1 reporter were very small compared to that of the later timepoint, which may be one of the reasons why *ret-1* was not identified in the sequencing-based analysis. Also, the sequencing was done on RNA extracted from the whole animals which may complicate the analysis and average out any tissue specific effects of *ret-1* splicing.

3) Figure 5—figure supplement 1 shows sequencing reads coverage for several genes. If the figure is only matching pairs for their paired end 2x75nt sequencing, which it should be for this type of analysis, the level of intronic signal for several genes is unexpected. So, when they write, "The splicing patterns of these genes are not significantly affected by *S. aureus* infection or the rnp-6 G281D mutation", what is the criterion for "significant", especially if one looks at the last example, cebp-1, for which there is a stark difference regarding retention of the last intron between wild-type and rnp-6 samples. [typo: *S. aurues* in figure]. This is particularly relevant given the known links between cebp-1, its 3' UTR (pubmed/28673818), pmk-1 (pubmed/27927209) and innate immunity.

q value/p adjusted < 0.05 is considered to be significant. The following are the additional replicates of the sequencing data of *cebp-1*. The retention of the last intron is variable between biological replicates. Thus, this is likely to be a noise instead of a true biological phenotype.

**Author response image 11. sa2fig11:** 

4) To understand the basis of altered resistance, it is standard in the field to perform simple tests to assay the extent of bacterial colonization.

We did not observe any observable effects of *rnp-6(dh1127)*, *rnp-6* RNAi or overexpression on *S. aureus* colonization, suggesting that *rnp-6* affects infection resistance through other mechanisms (see Figure 1—figure supplement 1I, Figure 4—figure supplement 1B and H).

5) Interpretation of Figure 2E is complicated by the fact that the constitutive expression level of at least one of the genes under study differs between wild-type and mutant worms. Thus, for nlp-30, there is the same fold-change after infection in mutant and wild-type, so the induction is not rnp-6 dependent. One therefore wonders whether their strategy for gene selection (Figure 2C) is entirely appropriate.

The relative levels of *nlp-30* are lower in *rnp-6(dh1127)* mutants regardless of infection, suggesting that the expression of *nlp-30* is dependent on *rnp-6* under both basal and infection conditions. *nlp-30* is still induced in infected *rnp-6(dh1127)* animals, suggesting that the mutation does not completely inhibit the induction of *nlp-30* by infection. The mutated RNP-6 may still mediate the induction of *nlp-30*. Alternatively, there may be alternative and/or compensatory mechanisms to regulate the expression of *nlp-30* in *rnp-6(dh1127)* animals. Nevertheless, the data supports that *rnp-6* affects the expression of *nlp-30*, and also additional regulation exists.

[Editors’ note: what follows is the authors’ response to the second round of review.]

Essential revisions:1) Despite the additional data provided, it is still not clear how RNP-6 impacts the PMK-1 pathway during *S. aureus* and *P. aeruginosa* infection. "RNP-6 exerts its inhibitory effects on immunity by suppressing the activity of PMK-1" (L 305-306) is over-stated as the authors do not have proof that rnp-6 impact the activation (phosphorylation) of PMK-1 during infection. Indeed:– The authors find no effect of rnp-6 OE strain or rnp-6 G281D mutant on the phospho-level of PMK-1 during *S. aureus* infection (rebuttal experiment).– The authors state that rnp-6 RNAi reduces P-PMK-1 during *S. aureus* infection, but this result is rather questionable considering the western-blot shown (rebuttal). There are huge differences in total protein levels as revealed by the loading control (actin), but these differences are not as strong in total PMK-1. This either suggest that 1) that the level of total PMK-1 varies, or 2) the experiment is not well conducted.– Figure 5—figure supplement 1C: control of total PMK-1 has not been provided. In addition, the quantification of this blot clearly indicates that there is a huge variation between experiments, and that P-PMK1 variation between conditions is not statistically significant.I don't think this affect the quality of the manuscript, but the author should be accurate in their interpretation, and the their conclusion modified accordingly (subsection “RNP-6 inhibits PMK-1 MAPK signaling”). Similarly, the text should be changed in the Discussion "How altering splicing can lead to activation of PMK-1.…".

We consulted other experts from the field concerning pathogen dependent PMK-1 phosphorylation. Although it is well known that *pmk-1* is required for resistance to bacterial pathogens in *C. elegans*, and mutations or treatments that induce phosphorylation of PMK-1 lead to enhanced survival (Cao and Aballay, 2016; Kim et al., 2004; Park et al., 2018; Pukkila-Worley et al., 2012; Sun et al., 2011; Xiao et al., 2020), whether bacterial pathogens actually induce phosphorylation of PMK-1 is debated.

*S. aureus* does not activate PMK-1 phosphorylation (Dr. Javier Irazoqui, personal communications), which we also confirmed. On the other hand, PA14 infection leads to high variability of phospho-PMK-1 levels (Dr. Read Pukkilla-Worley, personal communication), which is also consistent with our data. Conceivably these differences might reflect differences in timing, or culture conditions.

Therefore, the absence of PMK-1 phosphorylation upon infection is not inconsistent with previous findings, and does not compromise the results from survival epistasis and phosphoPMK-1 blots of *rnp-6i* treated animals, which indicate *pmk-1* acts downstream of *rnp-6*.

We have modified the Discussion section according to the reviewers’ suggestions.

2) As requested, the authors have performed bacterial lawn occupancy experiments and found it is increased in rnp-6 G281D mutant compared to WT, which still leaves a doubt on a potential lack of avoidance behavior of the rnp-6 mutant (rebuttal). Fortunately, the author also provide full lawn survival experiments as requested and found the rnp-6 G281D and rnp-6 OE strains to also display Esp compared to WT animals, indicating that these strains indeed have an immune defect. The authors keep describing this immune mechanism as "resistance" which I believe is not an appropriate term, especially in this new version of the manuscript where the authors, to answer reviewer 3, now assess the bacterial burden. Resistance, as defined by the capacity to limit bacterial burden, is opposed to tolerance, defined as ability to limit health impact of a given pathogen burden (see Schneider and Ayres Nat Rev Immunol 2008, Medzhitov, Schneider and Soares, Science 2012). In this type of survival assays, where animals are constantly exposed and fed with pathogenic bacteria, it is difficult to assess the resistance capacity. Assessing resistance capacity is easier to detect when animals are transfer onto non-pathogenic food (Sifri et al., Infect Immun 2003). Accordingly, the colonization assays, performed on animals constantly fed for 24h on *S. aureus*, show similar bacterial burden between control and rnp-6G281D mutant, rnp-6 OE strains or rnp-6 RNAi treated animals (Figure 1—figure supplement 1E, Figure 4—figure supplement 1B-H). Rather, these full lawn survival assays, controlled by similar colonization assays, allow to assess the tolerance capacity of the strains. Together, these results indicate that the immune function of rnp-6 relies on a tolerance mechanism. This should be indicated in the text which, I believe, would improve the quality of the article. However, the term resistance, which has been repeatedly found though-out the manuscript, is miss-used and should be replaced by i.e.: "decreased sensitivity" or "increased survival" and only when appropriate, by tolerance. Whether rnp-6 has also resistance immune branch should not be excluded, as it was difficult to assess in this experimental setting. Please also correct : "rnp-6 RNAi did not affect *S. aureus* bacterial burden, suggesting the resistance is caused by independent mechanisms".

We thank the reviewers for pointing out the difference between resistance and tolerance/survival. We have modified the manuscript by omitting the word pathogen resistance, as suggested.

3) There are still problems with the data presented in Figure 3 and in Figure 3—figure supplement 1. Indeed, there is no demonstration of a correlation between splicing as measured by PCR and using reporter genes. Thus, they neither "generate this type of reporter for at least one of the genes they identified as undergoing differential splicing", nor do they provide PCR evidence to support their observations with the ret-1 reporter. As mCherry and GFP have different half-lives, changes in fluorescence intensity under a pathological situation need not reflect a change in splicing. The magnitude of the change of GFP expression is considerably higher than that of mCherry; it seems that a change in splicing is not the most important effect here. Further, according to the dot plot, the GFP level for the vast majority of "Infected rnp-6" worms is below that for the majority of "rnp-6(dh1127)" worms, so the photomicrograph is totally unrepresentative and very misleading.

Because this reporter had been previously published as reflecting splicing (Heintz et al., 2017; Kuroyanagi et al., 2013), we took it at face value, but are grateful that the reviewers pointed out possible different interpretations of this reporter.

To address the reviewers’ concerns, we performed RT-PCR to assess the relative abundance of the two isoforms of *ret-1*. (see Figure 4C and D, Figure 4—figure supplement 1D).

First, we confirmed that both *rnp-6* RNAi and *S. aureus* infection induced *ret-1* exon 5

skipping, although the effects seem to be less dramatic as indicated by the reporter system, suggesting that the splicing reporter might not truly reflect the splicing of the endogenous *ret-1* transcript.

Furthermore, we observed that the *rnp-6(dh1127)* mutation did not inhibit the infection induced splicing of *ret-1*. Rather, the basal level of exon 5 skipping is slightly elevated in the *rnp-6* mutants. These observations were not reflected by the splicing reporter.

Multiple explanations could potentially account for the discrepancy of the two methods. First, we focused on a specific tissue (anterior gut) for the splicing reporter experiments whereas RT-PCR reflects the splicing events averaged over the whole animals. Second, some elements, such as splicing enhancers, may not be included into the reporter, which can lead to different splicing patterns compared to endogenous *ret-1* transcript. And finally, the signals from the reporter might reflect other processes, including mRNA and/or protein stability, translation and protein degradation.

We conclude that the *ret-1* splicing reporter does not accurately reflect the splicing of *ret-1* endogenous transcript. Therefore, we decided to remove the relevant data from the manuscript (Figure 3E and Figure 4C).

To replace Figure 4C, we checked how *rnp-6* RNAi affects the splicing (as measured by RTPCR) of *prg-2*, *gyg-1* and *tos-1*, which we identified to show *rnp-6* G281D dependence upon infection (Figure 3) (see Figure 4C and D and Figure 4—figure supplement 1D).

We observed that *rnp-6* RNAi affected *prg-2* and *tos-1* splicing, but had no observable effect on *gyg-1*. *rnp-6* RNAi promoted the splicing of *prg-2* intron 1, similar to the effect of *S. aureus* infection (Figure 3B), but opposite to *rnp-6 G281D* mutant. On the other hand, *rnp-6* RNAi inhibited the splicing of *tos-1* intron 1, and this splicing pattern was similar to what we observed for the *rnp-6* G281D mutant (Figure 3D).

In conclusion, the *rnp-6* G281D mutation inhibits infection induced *prg-2* intron 1 splicing, whereas *rnp-6* RNAi promotes the splicing event. In this regard, the *rnp-6* G281D mutation resembles a gain-of function. However, both *rnp-6* G281D and *rnp-6* RNAi reduce the splicing of *tos-1* intron 1. Therefore, the G281D mutation and *rnp-6* RNAi result in partially opposite and also overlap splicing, similar to the results of transcript abundance of immune-genes (Figure 4—figure supplement 1). Nevertheless, these results confirm that *rnp-6* RNAi affects splicing of the genes whose splicing is moderated by infection, strengthening the role of RNP-6 in infection induced splicing remodeling.

4) Statements like, "By contrast, we observed that the infection sensitive G281D substitution mutant of RNP-6 had a consistently suppressed phosphorylation levels of PMK-1 under *P. aeruginosa* infection condition in multiple biological replicates, although this did not reach statistical significance due variability of phospho-PMK-1 levels between replicates" are not appropriate. The results are simply not statistically significant and no inferences can be made.

As discussed, phospho-PMK-1 levels upon infection can be very variable. We removed this data from the manuscript.

References:

Cao, X., and Aballay, A. (2016). Neural Inhibition of Dopaminergic Signaling Enhances Immunity in a Cell-Non-autonomous Manner. Curr Biol, 26(17), 2329-2334. doi:10.1016/j.cub.2016.06.036

Kim, D. H., Liberati, N. T., Mizuno, T., Inoue, H., Hisamoto, N., Matsumoto, K., and Ausubel, F. M. (2004). Integration of *Caenorhabditis elegans* MAPK pathways mediating immunity and stress resistance by MEK-1 MAPK kinase and VHP-1 MAPK phosphatase. Proc Natl Acad Sci U S A, 101(30), 10990-10994. doi:10.1073/pnas.0403546101

Kuroyanagi, H., Watanabe, Y., Suzuki, Y., and Hagiwara, M. (2013). Position-dependent and neuron-specific splicing regulation by the CELF family RNA-binding protein UNC-75 in *Caenorhabditis elegans*. Nucleic Acids Res, 41(7), 4015-4025. doi:10.1093/nar/gkt097

Park, M. R., Ryu, S., Maburutse, B. E., Oh, N. S., Kim, S. H., Oh, S., . . . Kim, Y. (2018). Probiotic Lactobacillus fermentum strain JDFM216 stimulates the longevity and immune response of *Caenorhabditis elegans* through a nuclear hormone receptor. Sci Rep, 8(1), 7441. doi:10.1038/s41598-018-25333-8

Pukkila-Worley, R., Feinbaum, R., Kirienko, N. V., Larkins-Ford, J., Conery, A. L., and Ausubel, F. M. (2012). Stimulation of host immune defenses by a small molecule protects *C. elegans* from bacterial infection. PLoS Genet, 8(6), e1002733. doi:10.1371/journal.pgen.1002733

Xiao, Y., Liu, F., Li, S., Jiang, N., Yu, C., Zhu, X., . . . Liu, Y. (2020). Metformin promotes innate immunity through a conserved PMK-1/p38 MAPK pathway. Virulence, 11(1), 39-48. doi:10.1080/21505594.2019.1706305